# The Therapeutic Potential of Extracellular Vesicles in Psoriasis Treatment: Mechanisms, Applications, and Prospects

**DOI:** 10.3390/ijms262110297

**Published:** 2025-10-22

**Authors:** Ahmed Abdal Dayem, Myeongjin Song, Junhyeok Park, Ki-Heon Jeong, Kyung Min Lim, Sejong Kim, Kwonwoo Song, Ssang-Goo Cho

**Affiliations:** 1Stem Cell and Regenerative Biotechnology Major, School of Advanced Biotechnology, College of Institute of Science and Technology, Molecular & Cellular Reprogramming Center, Institute of Advanced Regenerative Science, Institute of Health, Aging & Society, Konkuk University, Seoul 05029, Republic of Korea; ahmed@konkuk.ac.kr (A.A.D.); bee060318@konkuk.ac.kr (M.S.); wnsgur4539@naver.com (J.P.); 2R&D Team, StemExOne Co., Ltd., 102, 19, Achasan-ro 5gil, Seongdong-gu, Seoul 04793, Republic of Korea; lmin0217@naver.com (K.M.L.); rlatpwhdc@gmail.com (S.K.); rnjsdnthd814@naver.com (K.S.); 3Department of Dermatology, College of Medicine, Kyung Hee University, Seoul 02447, Republic of Korea; khjeong246@khu.ac.kr

**Keywords:** extracellular vesicles, psoriasis, immunomodulation, signaling pathways

## Abstract

Psoriasis is a chronic inflammatory skin disease driven by dysregulated immune responses and aberrant keratinocyte (KC) proliferation, with a profound impact on patient quality of life. Emerging evidence highlights extracellular vesicles (EVs) as promising therapeutic candidates in regenerative medicine, offering new avenues for psoriasis management. This review provides a critical overview of psoriasis pathophysiology and evaluates the mechanistic basis of EV-based therapies, emphasizing their immunomodulatory capacity to restore immune homeostasis. We synthesize findings from preclinical studies, demonstrating the therapeutic potential of EVs derived from diverse cellular sources, including their ability to attenuate inflammation, regulate immune responses, enhance wound repair, and modulate KC function. Finally, we explore future directions aimed at optimizing EV therapeutic efficacy and translating these findings into clinical practice. Collectively, this review underscores EVs as a novel, targeted, and cell-free therapeutic strategy with the potential to transform psoriasis treatment.

## 1. Introduction

Psoriasis is a chronic inflammatory skin disorder affecting approximately 2–3% of the global population, characterized by chronic inflammation, rapid keratinocyte (KC) proliferation, and the infiltration of T-cells, neutrophils, dendritic cells, and macrophages [1]. The condition also involves significant skin barrier dysfunction [2,3]. It affects a significant portion of the population both in the United States and globally [4]. Data from the large-scale Multinational Assessment of Psoriasis and Psoriatic Arthritis (MAPP) surveys demonstrated that prevalence rates range from 1.4% in Spain to 3.3% in Canada, with a global average of 1.9% and 2.2% in the U.S. [5]. Specifically, the occurrence of psoriasis varies geographically and among ethnic groups within the same region, with notably higher prevalence in areas located at higher latitudes [2,6]. The disease is characterized by dysregulated immune responses, particularly involving T cells and other immune mediators, which drive the inflammatory cascade and the characteristic symptoms of psoriatic lesions, including erythema, scaling, and itching. The physical and psychological burden of psoriasis can severely impact patients’ quality of life, underscoring the urgent need for effective and innovative treatment strategies.

The causes of psoriasis are complex and diverse, including factors such as genetic factors, trauma, infections, drugs, sunlight, pregnancy, and psychogenic factors [1]. Furthermore, numerous epidemiological investigations have suggested a link between psychiatric conditions, including depression, anxiety disorders, obsessive–compulsive disorder, schizophrenia, and dementia [7,8,9,10,11].

Psoriatic plaques can be both disfiguring and accompanied by intense itching or pain, with pruritus often cited as the most distressing symptom [5]. The condition can significantly diminish quality of life, with many individuals experiencing profound social and psychological challenges in addition to physical discomfort [12,13]. The functional limitations associated with psoriasis are on par with, or even exceed, those experienced in primary health conditions such as cancer, cardiovascular disease, and depression [14]. At the cellular level, psoriasis involves complex interactions between innate and adaptive immune systems, particularly through the interleukin-23 (IL-23)/IL-17 axis, leading to aberrant KC proliferation and inflammatory cascade activation [15,16,17].

Psoriasis typically manifests as erythematous, scaly plaques on the scalp, trunk, and gluteal region [3]. Its pathogenesis is multifactorial, involving genetic predisposition, immune dysregulation, and environmental triggers, with the IL-23/IL-17 axis playing a pivotal role [18,19]. Dermal dendritic cells (dDCs) secrete IL-23 and IL-6 to drive T helper 17 (Th17) differentiation, and these Th17 cells release IL-17A/F and IL-22, promoting aberrant KC proliferation and differentiation [20,21]. Activated KCs further amplify inflammation by releasing tumor necrosis factor-alpha (TNF-α), IL-1β, IL-6, and LL-37-bound self-DNA/RNA to stimulate dendritic cells (DCs) [22,23]. Targeting the IL-23/IL-17 axis has yielded effective biologic therapies for psoriasis, yet clinical remission is often transient, with relapses common after the end of the therapeutic course [24,25]. Ongoing studies are therefore needed to elucidate the cellular and molecular interactions both within the immune system and between immune and non-immune cells that drive disease onset and recurrence.

Recent advancements in regenerative medicine have highlighted the therapeutic potential of extracellular vesicles (EVs) as a novel approach to managing psoriasis [26,27,28]. EVs are lipid-bilayer-enclosed particles released by various cell types, carrying bioactive molecules such as proteins, lipids, and nucleic acids that can modulate cellular communication and influence multiple biological processes, including inflammation and tissue repair [29,30]. The immunomodulatory properties of EVs have garnered attention for their ability to restore immune balance, potentially counteracting the hyperactivity observed in psoriasis [31,32].

Numerous in vitro and in vivo studies have demonstrated the efficacy of EVs derived from different cellular sources, including stem cells, mesenchymal stem cells (MSCs), and immune cells, in alleviating psoriatic symptoms [33,34,35]. These studies indicate that EVs can play essential roles in reducing inflammation, promoting wound healing, and regulating KC function, suggesting that they may serve as effective therapeutic agents in the treatment of psoriasis.

In this review, we organize the therapeutic use of EVs for psoriasis into three conceptual classes: natural immunomodulators, in which unmodified EVs leverage endogenous cargo (e.g., miRNAs, proteins, lipids) to attenuate inflammatory signaling and rebalance cutaneous immunity; engineered/modified EVs, genetically or chemically tailored to improve cell/tissue targeting, cargo loading, pharmacokinetics, or potency [36]; and drug-delivery systems, where EVs serve as carriers for exogenous therapeutics (e.g., small molecules, siRNAs/ASOs, biologics) to enable targeted, cell-selective delivery with reduced systemic exposure [37]. This framework clarifies mechanisms, design considerations, and translational paths for EV-based interventions in psoriasis.

Despite their promise, several challenges remain regarding the clinical application of EVs in psoriasis treatment. Issues related to isolating, characterizing, and standardizing EVs must be addressed to ensure their safety, efficacy, and reproducibility in therapeutic contexts. Furthermore, optimized delivery methods enhance EVs’ bioavailability and therapeutic effects in target tissues.

This review critically examines the current research landscape on the mechanisms by which EVs affect psoriasis treatment. We will explore the immunomodulatory functions of EVs, summarize key findings from in vitro and in vivo studies, and address the limitations and challenges associated with their clinical applications. Finally, we will outline future directions for enhancing the therapeutic efficacy of EVs in psoriasis, providing a comprehensive overview of their potential as a novel therapeutic strategy to improve patient outcomes.

## 2. Overview of Psoriasis Pathophysiology

Psoriasis, a chronic inflammatory disease, is characterized by abnormal proliferation of KCs due to excessive immune activity [38]. It affects over 60 million adults and children worldwide (WHO, Global Report on Psoriasis. Geneva: World Health Organization, 2016). The prevalence of psoriasis varies geographically, ranging from 0.1% in East Asia to 1.5% in Western Europe and 1.88% in Australia [39,40]. While the incidence differs among ethnic groups, psoriasis affects men and women equally [41]. In contrast to atopic dermatitis, psoriasis has a lower incidence in young children, with prevalence increasing with age [42].

The development of psoriasis is intricately linked to a combination of genetic predisposition, environmental factors, and immune system dysregulation. A dynamic interplay exists between immune cells and non-immune cells in response to these external and internal stimuli [43]. EVs are secreted by a variety of cell types, including DCs, macrophages, neutrophils, mast cells, T lymphocytes, KCs, and adipocytes, all of which have been implicated in the pathogenesis of psoriasis. Emerging research indicates that EVs produced from both immune and non-immune cells not only influence immune regulation but also possess the capacity to elicit anti-inflammatory effects [44].

Psoriasis is driven by a pathophysiological process characterized by excessive epidermal cell proliferation and disrupted KCs maturation. Its development is influenced by a multifaceted interplay of genetic predispositions, immune system abnormalities, and environmental factors [45]. Several medications have been linked to the onset or worsening of psoriasis, including angiotensin-converting enzyme (ACE) inhibitors, angiotensin II receptor blockers (ARBs), lithium, antimalarial agents, and nonsteroidal anti-inflammatory drugs (NSAIDs) [46,47,48]. Although ACE inhibitors and ARBs have been associated with psoriasis, a population-based case–control study reported no significant correlation with beta-blockers or other antihypertensive drugs [49]. Additional triggers may include infections, such as HIV or streptococcal infections, and physical trauma to the skin, known as the Koebner phenomenon [46,50].

A multifaceted interaction between genetic predisposition and immune system dysregulation drives psoriasis pathogenesis. Comparative transcriptomic analyses have revealed thousands of differentially expressed genes between lesional, non-lesional, and healthy skin, underscoring the disease molecular complexity [51]. Psoriasis is widely recognized as a chronic inflammatory condition marked by aberrant immune activation. Research has demonstrated that the disease is associated with the activation of DCs and T lymphocytes, which, in turn, stimulate the production of key pro-inflammatory cytokines, including IL-17, IL-23, and TNF-α [52,53]. These cytokines promote hyperproliferation of KCs, impaired differentiation, and an inflammatory milieu in the dermis, contributing to hallmark features such as erythema and scaling [54,55,56,57].

Psoriasis can be classified clinically into several subtypes, including plaque, erythrodermic, pustular, inverse (flexural), and guttate psoriasis [40]. Plaque psoriasis is the most common form, accounting for approximately 90% of cases. It presents as well-demarcated papulosquamous plaques that are distinctly separated from surrounding healthy skin, typically appearing on the elbows, knees, scalp, lumbosacral region, and umbilicus [58]. Erythrodermic psoriasis involves about 80% of the body surface and is characterized by widespread erythematous lesions [40]. Guttate psoriasis begins as small, droplet-shaped lesions and is often triggered by streptococcal infections [59]. Based on the histological analysis, psoriasis is characterized by hyperkeratosis, marked by excessive proliferation of KCs in the stratum corneum; parakeratosis, the presence of KCs retaining their nuclei within the stratum corneum; and acanthosis, the thickening of the epidermal layer [4].

The genetic studies identified the psoriasis in the susceptibility-1 locus (PSORS1) region within the major histocompatibility complex (MHC) as a key susceptibility locus for psoriasis, with Human leukocyte antigen (HLA)-Cw6 later identified as the primary associated allele [60]. More recent genome-wide association studies have uncovered 63 genes linked to roughly 28% of psoriasis heritability, many of which are involved in inflammatory and immune pathways [61]. Notably, Single Nucleotide Polymorphism (SNP) analyses have revealed connections between psoriasis and genes related to Th2 (e.g., IL-4, IL-13), Th17 (e.g., IL-12, IL-23), innate immune responses (e.g., Nuclear factor kappa-light-chain-enhancer of activated B cells (NF-κB), interferons (IFNs), and cluster of differentiation 8 (CD8)^+^ T cell-mediated adaptive immunity [62].

Psoriasis was initially believed to be a KC-driven disorder; however, recent studies have revealed its strong association with immune dysregulation. KCs are pivotal in the initiation and persistence of psoriasis (Figure 1). As innate immune cells, they respond to diverse triggers, and when stressed, release nucleotides and antimicrobial peptides that activate plasmacytoid dendritic cells, leading to maturation of myeloid dendritic cells via IFN-α, IFN-γ, TNF-α, and IL-1β [43,63]. Beyond initiating disease, keratinocytes amplify inflammation during the maintenance phase [64]. Stimulated by proinflammatory cytokines, they proliferate excessively and secrete chemokines (CXCL1/2/3, CXCL8, CXCL9/10/11, CCL2, CCL20) that recruit immune cells, antimicrobial peptides (S100A7/8/9/12, hBD2, LL37), and other mediators that intensify the immune response. Together with fibroblasts and endothelial cells, they promote vascular activation and extracellular matrix remodeling [3,17,65]. Continuous crosstalk with Th17 cells sustains psoriatic pathology, driving keratinocyte hyperproliferation, abnormal differentiation, vascular changes, and leukocyte infiltration [64,66,67].

Both extrinsic and intrinsic factors influence psoriasis. Extrinsic triggers include infections, skin trauma, lifestyle factors, medications, humidity, cold weather, and air pollution. The inherent factors involve microbiota dysbiosis, stress, dysregulated lipid metabolism, and imbalances in sex hormones [69]. These triggers lead to the activation of plasmacytoid dendritic cells (pDCs) [70] and dDCs, which produce IL-23, thereby promoting the polarization and clonal expansion of Th17 cells [17]. Additionally, activated pDCs and macrophages release pro-inflammatory cytokines such as IFN-α and TNF-α, further amplifying the inflammatory cascade in psoriasis pathogenesis [71].

The psoriasis-mediated immune reaction is mediated through the production of IFN-I, pDCs initiate protective immunity by activating classical dendritic cells (cDCs), T cells, natural killer (NK) cells, and B cells [70]. Research report suggests that IFN-α plays a more crucial role in the onset of psoriasis rather than its maintenance [72].

TNF-α, released during this phase, stimulates KCs to express pro-inflammatory cytokines, including Intercellular Adhesion Molecule-1 (ICAM-1), C-X-C motif chemokine ligand 8 (CXCL8), IL-1β, and IL-6 [71]. IFN-α and TNF-α, secreted by pDCs, macrophages, and other immune cells, contribute to the maturation and activation of myeloid dendritic cells (mDCs), which play a pivotal role in chronic psoriasis [71]. mDCs and dDCs exhibit distinct functions in psoriasis pathogenesis, underscoring the need for further investigation into their specific contributions. EVs released by cytokine-primed keratinocytes make a substantive contribution to psoriasis pathogenesis [57]. Acting as a previously underappreciated communication conduit between keratinocytes and neutrophils, these “psoriatic” EVs, enriched for a distinctive proteomic cargo, are internalized by neutrophils and activate NF-κB and p38 MAPK signaling, thereby augmenting NETosis and inducing pro-inflammatory cytokines (IL-6, IL-8, TNF-α). Collectively, the data delineate a keratinocyte–neutrophil EVs axis that amplifies cutaneous inflammation and highlight keratinocyte-derived EVs or their specific cargo, as actionable therapeutic targets for biogenesis/uptake blockade, cargo neutralization, or downstream pathway inhibition [57].

Neutrophils are a predominant immune infiltrate in psoriatic skin and are recruited in response to TNF-α and IL-17 secreted by KCs. This process induces KCs to release CXCL1, CXCL2, CXCL3, CXCL5, and CXCL8 (IL-8), further attracting neutrophils to psoriatic lesions [73]. Activated neutrophils contribute to inflammation by releasing reactive oxygen species (ROS), IL-17, LL37/cathelicidin, and neutrophil extracellular traps (NETs) [74]. Macrophages undergo polarization into the pro-inflammatory M1 phenotype in response to IFN-γ and TNF-α, leading to the production of IL-1β, IL-6, IL-12, IL-23, Monocyte Chemoattractant Protein-1 (MCP-1), and TNF-α [75]. When IL-23 is introduced into mouse peritoneal macrophages, they produce elevated levels of IL-17, IL-23, and IFN-γ, reinforcing the inflammatory loop [76]. In a psoriasis-like animal dermatitis model, UC-MSC therapy was safe and efficacious [77]. Among delivery routes, subcutaneous administration proved optimal, avoiding the elevated mortality observed with intravenous injection. Mechanistically, umbilical cord mesenchymal stem cells (UC-MSCs) attenuated cutaneous inflammation predominantly by suppressing IL-17-producing γδ T cells. Collectively, these data provide mechanistic insight and support subcutaneous UC-MSC administration as a promising therapeutic strategy for psoriasis. [77].

IL-23 links innate immunity (by stimulating NK cells and neutrophils to produce IL-17) with adaptive immunity (by promoting IL-17 production from T cells) [78]. IL-23 is crucial for maintaining the Th17 lineage, and in its absence, Th17 cells lose their phenotype [79]. It is secreted by antigen-presenting Langerhans cells [80], DCs [81], monocytes, and macrophages [82]. Upon binding to its receptor, IL-23 initiates The Janus kinase(JAK)2-signal transducer and activator of transcription(STAT)3 phosphorylation [83], leading to increased IL-17 production and activation of the transcription factor retinoid-related orphan receptor (ROR)-γt [84]. Th17 cells secrete a range of cytokines, including IL-17, IL-21, IL-22, IL-26, and TNF-α, all of which contribute to psoriatic inflammation [85]. Stimulation of human epidermal sheets with IL-17A increases the expression of IL-22 and IFN-γ [86]. Compared with monoculture, psoriasis-derived mesenchymal stromal cells (PsO-MSCs) co-cultured with healthy donor MSCs (H-MSCs) secreted substantially lower levels of hallmark psoriatic pro-inflammatory cytokines (IL-6, IL-12, IL-13, IL-17A, TNF, and G-CSF) [87]. In contrast, IL-2 secretion increased toward levels observed in H-MSCs. Collectively, these findings indicate that indirect co-culture with H-MSCs restores a more physiological psoriasis-derived MSCs phenotype via paracrine signaling, as evidenced by attenuation of hyperproliferation and re-balancing of the Th1/Th17–Th2 cytokine axis [87]. An allogeneic umbilical cord–derived MSCs transplantation study reported an acceptable safety profile and partial clinical efficacy in patients with psoriasis [88]. Mechanistically, MSCs appear to exert immunoregulatory effects by promoting the differentiation of naïve T cells into effector and memory subsets and, importantly, by rebalancing the Treg–Th17 axis—expanding regulatory T cells while reducing Th17 cells and serum IL-17 levels. Psoriatic KCs drive disease propagation by exporting miR-381-3p to CD4^+^ T cells via EVs [89]. Vesicle-delivered miR-381-3p skews T-cell differentiation toward Th1 and Th17 lineages by post-transcriptionally repressing the negative immune regulators UBR5 and FOXO1 [89]. These data delineate a KC-derived EV T-cell axis that amplifies pathogenic immunity in psoriasis and nominates KC-derived EVs or their specific cargo, miR-381-3p, as tractable therapeutic targets for cargo inhibition, vesicle blockade, or selective cargo editing.

IL-17 also enhances the expression of the human cathelicidin antimicrobial peptide LL37, an autoantigenic antimicrobial peptide, and CXCL1, both of which promote the inflammatory response in psoriasis [90]. CXCL1 further upregulates a disintegrin-like and metalloprotease domain-containing thrombospondin type 1 motif-like 5 (ADAMTSL5), which, in turn, amplifies IL-17A and IFN-γ expression, exacerbating inflammation [91].

## 3. The Therapeutic Applications of Mesenchymal Stem Cells

MSCs are multipotent stem cells characterized by their self-renewal capacity and immunomodulatory properties [92]. The ability of MSCs to regulate immune responses, particularly through interactions with immune cells, is well-established. Accordingly, MSCs represent a promising therapeutic alternative for treating immune-mediated disorders such as psoriasis [93].

In psoriasis, excessive activation of Th1 and Th17 immune cells plays a central role in disease pathogenesis. MSCs have been shown to suppress the activity of both Th1 and Th17 cells while promoting Th2 cell activation to restore immune balance [94]. Although Th2 cells confer potential benefits by secreting anti-inflammatory cytokines such as IL-4 and IL-13 [95], their overactivation may lead to other immune-related conditions, including atopic dermatitis [96]. Inactivation of Th1 and Th17 cells led to a marked reduction in the secretion of their key cytokines, IL-17 and IL-23. Additionally, pro-inflammatory cytokines, including IFN-γ, IFN-I, and TNF-α, were significantly downregulated [94,97]. Beyond immunosuppression, MSCs also upregulate the anti-inflammatory cytokines IL-10 and transforming growth factor beta (TGF-β) during psoriasis treatment [94,98]. Owing to MSC-mediated suppression of Th1 and Th17 cells, the secretion of their signature cytokines, IL-17 and IFN-γ, was significantly reduced. Furthermore, the expression of additional pro-inflammatory cytokines, including IL-22 and TNF-α, was also downregulated [88,94]. IL-10 exerts its anti-inflammatory effects by modulating key signaling pathways, including JAK/STAT, NF-κB, and extracellular signal-regulated kinase 1 and 2 (ERK1/2), a mitogen-activated protein kinase (MAPK) [99,100]. In contrast, TGF-β mediates its effects primarily through Smad and NF-κB signaling, contributing to the alleviation of psoriatic inflammation [101,102].

Neutrophil infiltration is also a key contributor to the pathogenesis of psoriasis. This process involves neutrophils migrating from the bloodstream into the skin tissue, where they induce inflammatory responses [103]. MSCs have been shown to reduce neutrophil infiltration, including subsets such as CD3^+^ CD4^+^ CD11b^+^, CD11c^+^, Lymphocyte antigen 6G-positive (Ly6G^+^) cells, by modulating CXCL chemokine signaling pathways [98,104,105,106].

Furthermore, MSCs suppress the activity of specific enzymes involved in neutrophil-mediated inflammation. Platelets play a role in facilitating neutrophil migration by interacting with neutrophils, and this interaction is promoted through the action of the antimicrobial peptide cathelicidin-related antimicrobial peptide (CRAMP). MSCs can inhibit CRAMP, thereby reduce platelet activation and prevent their interaction with neutrophils [107,108]. In addition, myeloperoxidase (MPO), an enzyme that regulates neutrophil degranulation and NET formation, has been implicated in disease severity. The inhibition of MPO has been shown to attenuate psoriatic inflammation and improve clinical symptoms [109,110].

MSCs inherently possess the capacity to alleviate psoriasis; however, this therapeutic effect can be further enhanced through a process known as “licensing,” in which MSCs are preconditioned with pro-inflammatory cytokines to augment their anti-inflammatory secretory profile [111]. When MSCs are pretreated with IFN-γ and TNF-α, they exhibit increased secretion of immunomodulatory factors, including tumor necrosis factor-stimulated gene-6 (TSG-6), indoleamine 2,3-dioxygenase (IDO), cyclooxygenase-2 (COX-2), prostaglandin E2 (PGE2), and IL-10. This enhanced response suppresses the phosphorylation of STAT1, leading to reduced expression of CXCL1 and ultimately contributing to a more effective reduction in inflammation and neutrophil infiltration [88,105].

In clinical settings, MSC-based therapies have shown promising outcomes in treating psoriasis, with minimal adverse effects such as mild fever [88]. Notably, patients demonstrated significant improvements in disease severity, with sustained therapeutic effects lasting up to one-year post-treatment [88,112]. While MSCs hold great promise as a therapeutic alternative for psoriasis, challenges remain, including limited clinical data, high production costs, and concerns about their longevity and therapeutic consistency [19].

## 4. Mesenchymal Stem Cell-Derived Conditioned Media and Psoriasis Therapy

MSC-derived conditioned medium (MSC-CM), a complex mixture of secreted bioactive factors including EVs, cytokines, and growth factors, has emerged as a promising cell-free therapeutic alternative for inflammatory skin disorders, including psoriasis. This approach bypasses the potential risks of direct stem cell transplantation, such as tumorigenicity and immunogenicity, while preserving MSCs’ immunomodulatory and regenerative properties.

The transplantation of MSCs at injury sites is often hindered by the dense, complex structure of surrounding tissues, which can lead to rapid cell death or clearance by local immune cells [113]. Extensive research has focused on understanding the mechanisms by which MSCs mediate their regenerative and immunomodulatory effects. In models of skin injury, MSCs-CM have been shown to enhance KCs and endothelial cell activity, while also promoting macrophage recruitment to facilitate wound healing [114]. Given the prominent role of MSCs paracrine signaling in tissue regeneration, there is growing interest in deciphering the molecular composition and functional significance of the secreted factors that mediate interactions between MSCs and the host tissue microenvironment.

It has been widely recognized that the therapeutic benefits of MSCs in treating skin disorders are primarily attributed to their ability to migrate to damaged tissue and exert immunomodulatory, anti-autoimmune, and paracrine effects. Preclinical studies have also demonstrated that conditioned medium (CM) obtained from stem cell culture can effectively promote healing in psoriasis-like lesions, positioning CM as a promising alternative to direct cell-based therapies [115,116]. These regenerative effects are mainly driven by bioactive paracrine molecules, including cytokines, chemokines, and growth factors, secreted by abundant stem cells in CM or culture supernatant [114,117]. Given its regenerative potential, CM represents an innovative therapeutic strategy in regenerative medicine, with several preclinical models demonstrating favorable outcomes.

A study conducted by our research team focused on formulating a safe, user-friendly emulsion cream for psoriasis, using MSCs-CM enriched with EVs [38]. To enhance EV yield, a modified three-dimensional (3D) culture system incorporating TGF-β was developed for Wharton’s jelly mesenchymal stem cells (WJ-MSCs). The 3D culture approach significantly increased EV production compared to traditional two-dimensional (2D) methods. CM obtained from this system was incorporated into a topical cream, and its therapeutic efficacy was evaluated in a mouse model of imiquimod (IMQ)-induced psoriasiform dermatitis [38]. Application of the EV-enriched 3D-WJ-MSCs-CM cream markedly reduced key clinical features of psoriasis—erythema, epidermal thickening, and scaling—while also reversing associated histopathological abnormalities. Furthermore, elevated mRNA levels of pro-inflammatory cytokines, including IL-17a, IL-22, IL-23, and IL-36, were substantially downregulated in treated lesions [38]. In sum, this study introduces a novel MSC-derived EV-enriched topical formulation as a promising therapeutic strategy for the management of psoriasis.

A study discusses the case of a 47-year-old Indonesian male patient diagnosed with moderate psoriasis vulgaris who had not received standard treatment. At the beginning of the treatment period, the patient had a Psoriasis Area Severity Index (PASI) score of 10.8. His condition included large plaque lesions covering almost 20% of his body surface area, negatively impacting his quality of life. The lesions featured thick silvery scales and erythema, particularly in interphalangeal joints and lower limbs.

A case study evaluated the effectiveness of MSCs-CM derived from WJ-MSCs in treating patients with Psoriasis Vulgaris [118]. MSC-CM was collected at passage 8 from conditioned media and processed by centrifugation and tangential flow filtration (TFF) using 2–1000 kDa filter cassettes. The patient received a 7-day treatment regimen consisting of an initial two cc intravenous infusion of MSCs-CM, followed by four consecutive daily intramuscular injections of 1.3 cc targeting clinically active psoriatic lesions [118]. Additionally, the patient applied a topical gel containing MSCs-CM daily to the affected areas and was closely monitored for adverse effects. Clinical improvement was evident within one week. The PASI score decreased from 10.8 to 3.2, with reductions in visible erythema, scaling, and induration. By the end of treatment, approximately 60% clearance of psoriatic plaques was achieved, along with marked improvement in skin texture [118].

This case study highlights the therapeutic potential of MSCs-CM as a safe and effective cell-free approach for treating moderate psoriasis vulgaris. It provides encouraging preliminary evidence supporting the use of stem cell-derived CM as an alternative to conventional cell-based therapies.

Another clinical study showcased the case of a 38-year-old male with a two-year history of psoriasis vulgaris, characterized by extensive erythematous plaques and silvery scales across the scalp, including the retroauricular areas [119]. The initial Psoriasis Scalp Severity Index (PSSI) score was 28. In this study, MSCs-CM was prepared from MSCs isolated from adipose tissue of a healthy donor. The treatment scheme involved daily topical application of MSCs-CM to the affected scalp for 1 month. The patient received no concurrent therapies during treatment or the six-month follow-up. Clinical evaluations were performed at regular intervals [119]. Within two weeks, a marked reduction in scaling was observed. After one-month, psoriatic plaques and silvery scales had resolved entirely, with the PSSI score decreasing from 28 to 4. No adverse effects were reported during treatment or follow-up.

The observed effects are attributed to the rich composition of bioactive factors, such as cytokines and growth factors, present in MSCs-CM, which may exert direct regenerative and immunomodulatory effects on skin tissue. Unlike direct MSC transplantation, MSCs-CM offers a cell-free alternative that bypasses issues related to cell survival and host integration.

Despite the promising results, this study acknowledges limitations. The precise identity and optimal combination of active molecules within MSCs-CM remain undefined. Further research is warranted to characterize these factors, elucidate the cellular and histological changes induced, and assess long-term therapeutic outcomes. Additionally, clinical data supporting the therapeutic use of CM remains scarce. Eventually, well-designed randomized controlled trials with larger cohorts are required to establish the long-term safety and clinical efficacy of CM-based treatments for psoriasis.

## 5. Psoriasis Key Signaling Pathways

### 5.1. JAK Signaling and Psoriasis

The JAK/STAT signaling cascade serves as a crucial conduit linking extracellular cytokine signals to changes in gene expression within the nucleus. This pathway is composed of four JAKs, including JAK1, JAK2, JAK3, and tyrosine kinase2 (TYK2), and seven STAT factors, including STAT1 through STAT6, with STAT5 existing as two isoforms such as STAT5A and STAT5B [120].

It plays a fundamental role in regulating various immune-related cellular functions, including proliferation, differentiation, apoptosis, and migration [121]. Upon engagement of cytokines with their respective receptors, associated JAKs are activated via transphosphorylation, which subsequently leads to the phosphorylation of STAT proteins. These activated STATs then dimerize and translocate into the nucleus, where they modulate the expression of target genes that govern immune regulation and inflammatory responses [122]. The fidelity of this signaling cascade is critical for immune system homeostasis, as its dysregulation, whether through hyperactivation or inadequate signaling, has been implicated in the pathogenesis of autoimmune conditions, persistent inflammatory states, and even cancer [123].

In psoriasis diseases, aberrant activation of the JAK/STAT signaling cascade plays a crucial role in driving persistent cytokine production and orchestrating localized immune responses within tissues. Among the STAT family, STAT3 emerges as a key transcription factor involved in the pathophysiology of these conditions. It is activated by several cytokines, including IL-6, IL-12, IL-21, and IL-2, but is particularly significant due to its ability to promote Th17 cell differentiation and stimulate IL-22 secretion processes that are central to KCs hyperproliferation and abnormal epidermal differentiation seen in psoriatic lesions [121,124].

Experimental data have shown that selective overexpression of STAT3 in KCs in mouse models is sufficient to induce spontaneous psoriasis-like skin changes [123]. Notably, KCs themselves can produce inflammatory cytokines such as IL-23 and IL-17E, thereby reinforcing inflammation via autocrine and paracrine mechanisms mediated by STAT3 and STAT5 activation [125]. These insights highlight STAT3 signaling in KCs as a pivotal driver of a self-sustaining inflammatory cycle that enhances the expression of antimicrobial peptides and various pro-inflammatory cytokines [126].

Furthermore, specific JAK family members contribute uniquely to distinct cytokine-mediated signaling pathways. For instance, TYK2 and JAK2 are central to IL-12 and IL-23 signaling and are essential for maintaining Th1 and Th17 immune responses [14]. TYK2 has drawn particular interest due to its role in facilitating STAT3 activation downstream of IL-23 signaling [121,127]. Therefore, the JAK/STAT axis functions not only in regulating immune cell interactions but also in modulating epithelial responses, underscoring its multifaceted contribution to the development and persistence of psoriatic disease.

Given its pivotal role in orchestrating immune responses, the JAK/STAT signaling axis has emerged as a prominent therapeutic target. Various JAK inhibitors, such as tofacitinib (TFC), baricitinib, and Upadacitinib, have been explored for their potential to treat psoriasis, demonstrating encouraging results in alleviating cutaneous and joint inflammation [122]. These agents function by blocking JAK-dependent phosphorylation steps, thereby inhibiting subsequent STAT activation and the transcription of pro-inflammatory genes. First-generation JAK inhibitors generally target multiple JAK isoforms concurrently, which can yield broad immunosuppressive effects. While this may enhance efficacy, it also increases the risk of adverse events due to widespread immune suppression [128]. To address this concern, newer therapeutic approaches are being developed to selectively inhibit specific JAK family members, such as TYK2, to achieve clinical benefits with fewer side effects [121]. The advancement of JAK/STAT-targeted drugs represents a strategic shift toward modulating intracellular signaling pathways to manage multifactorial immune-mediated disorders like psoriasis.

Among these, IL-23 acts as a crucial upstream cytokine, regulating the activity of IL-17 and IL-22, two key mediators associated with abnormal KCs proliferation and plaque development. In addition to activating IL-17 pathways, IL-23 has been implicated in inducing epigenetic modifications, such as dimethylation of histone H3 at lysine 9 (H3K9), potentially sustaining pathogenic inflammatory responses [68]. The transcription factor STAT3 intensifies this inflammatory cascade by upregulating IL-17E expression through interleukin-17 receptor B, prompting KCs to release various pro-inflammatory cytokines and chemokines [68]. IL-17A also contributes to disease progression by stimulating the transcription of numerous inflammatory mediators and recruiting immune cells into the epidermis. Although IL-22 is traditionally linked to tissue repair and epithelial regeneration, it promotes KC proliferation by inducing matrix metalloproteinases and anti-apoptotic molecules [129]. Additionally, IL-9 plays a role in sustaining the inflammatory environment by enhancing KCs’ growth and stimulating the release of helper T-cell-related cytokines and vascular endothelial growth factor (VEGF) [130]. Altogether, these disrupted cytokine interactions form a complex network that drives both the cutaneous and systemic symptoms seen in psoriasis.

Therapeutic strategies targeting the JAK/STAT signaling axis have demonstrated clinical efficacy in the management of psoriasis. Among the most extensively studied JAK inhibitors are TFC, ruxolitinib, and baricitinib, each of which has the potential to attenuate the inflammatory pathways central to disease pathogenesis. TFC, an oral inhibitor of JAK1 and JAK3, has undergone rigorous clinical evaluation. In a Phase IIb dose-ranging trial, administration of 2 mg, 5 mg, and 15 mg twice daily led to significant improvements in PASI scores compared to placebo, with the 5 mg and 15 mg groups achieving the highest PASI75 response rates. However, the 15 mg dose was associated with a higher incidence of adverse effects, suggesting that 5 mg twice daily offers an optimal balance between efficacy and safety [131].

The therapeutic effectiveness of JAK inhibitors in managing psoriasis primarily stems from their capacity to disrupt the JAK/STAT signaling pathway, which plays a pivotal role in mediating pro-inflammatory cytokine signals that contribute to KCs hyperproliferation and sustained inflammation [68]. Psoriasis is marked by dysregulated cytokine activity, particularly IL-23, IL-17, and IL-22, that not only intensifies immune activation but also impairs normal KCs differentiation and promotes excessive epidermal growth [132,133,134]. By selectively inhibiting JAK family kinases, especially JAK1, JAK2, and JAK3, these agents effectively suppress STAT-mediated transcription of genes associated with inflammation and KCs proliferation [68]. Both experimental and clinical studies have demonstrated that JAK inhibition attenuates cytokine-driven signaling, leading to reduced inflammatory responses in psoriatic lesions and restoration of normal epidermal dynamics. This combined anti-inflammatory and antiproliferative mechanism positions JAK inhibitors as a distinctive and promising class of therapeutics that target both immunologic and structural components of psoriatic pathology. Although the availability of both oral and topical formulations enhances their clinical utility, ongoing research is essential to establish long-term safety profiles, therapeutic efficacy, and optimal patient selection criteria.

### 5.2. NF-κB Signaling and Psoriasis

NF-κB plays a crucial role in modulating cell survival, activation, and the differentiation of innate immune cells and pro-inflammatory T cells [135]. Notably, it is a key regulator in host immune and inflammatory responses [136].

In mammals, the NF-κB transcription factor family comprises five members: p65 (RelA), RelB, c-Rel, p105/p50 (NF-κB1), and p100/p52 (NF-κB2). These proteins form various homo- and heterodimeric complexes with distinct transcriptional activities [137]. Under resting conditions, NF-κB dimers are sequestered in the cytoplasm by inhibitor of κB (IκB) proteins, including IκB-α, IκB-β, IκB-γ, IκB-ε, and Bcl-3. Upon activation, IκB proteins are phosphorylated by the IκB kinase (IKK) complex, leading to their ubiquitination and subsequent proteasomal degradation. This degradation allows NF-κB to be released and translocated into the nucleus, where it initiates transcription of target genes. Although NF-κB signaling is not essential for epidermal differentiation, the absence of IKKα prevents proper differentiation into the stratum corneum, indicating its critical role in terminal KCs maturation [138].

Epidermal appendages such as hair, teeth, nails, sweat glands, and sebaceous glands originate from the epidermis during ectodermal morphogenesis. During this process, NF-κB plays a pivotal role downstream of the ectodysplasin A receptor (EDAR), a key regulator of ectodermal development [139]. In addition to its well-established function in inflammation, NF-κB is involved in cutaneous immune responses. It regulates the transcription of several pro-inflammatory cytokines and is tightly associated with inflammatory processes in the skin [139].

At the cellular level, NF-κB functions as a master regulator of inflammatory responses, immune cell activation, and cell survival [135]. Upon nuclear translocation, NF-κB dimers bind to specific DNA sequences called κB sites in the promoter regions of target genes [140], thereby regulating the transcription of over 500 genes involved in inflammation, immunity, cell proliferation, and apoptosis [141]. NF-κB controls the expression of numerous pro-inflammatory cytokines, including TNF-α, IL-1β, IL-6, IL-8, and IL-12, as well as chemokines such as C-C motif chemokine ligand 2 (CCL2), CCL5, and CXCL10 [142]. Additionally, NF-κB regulates the expression of anti-apoptotic proteins like B-cell lymphoma 2 (Bcl-2), B-cell lymphoma-extra-large (Bcl-xL), thereby promoting cell survival and proliferation [143]. In immune cells, NF-κB is essential for the activation and differentiation of T cells, B cells, DCs, and macrophages [144]. It plays a crucial role in the development of Th1 and Th17 cell responses, which are central to the pathogenesis of autoimmune and inflammatory diseases.

Although NF-κB signaling is not essential for epidermal differentiation, the absence of IKKα prevents proper differentiation into the stratum corneum, indicating its critical role in terminal KCs maturation. In psoriasis pathogenesis, NF-κB activation serves as a central hub orchestrating chronic inflammation and abnormal KCs proliferation [145]. Studies have demonstrated that NF-κB is constitutively activated in psoriatic lesions, with intense nuclear staining observed in both KCs and infiltrating immune cells [146]. The activation of NF-κB in psoriatic epidermis correlates directly with disease severity, as epidermal nuclear positivity for NF-κB shows significant correlation with the grade of epidermal hyperplasia [147]. As previously mentioned, IL-23 activates Th17 cells, leading to the production of IL-17A, IL-17F, and TNF [148]. IL-17 further induces downstream gene expression by activating canonical NF-κB signaling, the CCAAT/enhancer-binding protein (C/EBP) family, and MAPK pathways [20].

NF-κB plays a pivotal role in connecting KCs dysfunction with immune cell activation in psoriatic lesions [149]. Inflammatory cytokines such as TNF-α, IL-17, and IL-22 activate NF-κB in KCs, leading to the production of additional pro-inflammatory mediators, chemokines, and antimicrobial peptides [150,151]. This creates a self-sustaining inflammatory loop where activated KCs further recruit and activate immune cells [152].

Experimental evidence from various mouse models has definitively established the critical role of NF-κB in psoriasis pathogenesis [149]. Mice with global deletion of IκBα develop spontaneous psoriasis-like skin symptoms, while IκBα deficiency specifically in KCs results in epidermal hyperplasia without significant epidermal inflammation [153]. Importantly, loss of IκBα in both KCs and T cells leads to a phenotype similar to global deficiency, demonstrating that NF-κB activation in both cell types is essential for psoriasis development [154]. Conversely, KC-specific deletion of RelA (p65) rescues the phenotype observed in global IκBα knockout mice [145]. In the CD18 hypomorphic (CD18hypo) mouse model of psoriasis, treatment with the IκB kinase inhibitor acetyl-11-keto-β-boswellic acid (AKβBA) profoundly suppressed NF-κB signaling and subsequent cytokine production [155].

Mice deficient in NF-κB showed alleviated IMQ-induced psoriasis-like dermatitis [156]. Furthermore, intradermal injection of plasmin, which activates NF-κB signaling and induces the production of inflammatory factors, including CCL20 and IL-23, results in psoriasiform skin inflammation with several features of human psoriasis in mice [157].

Clinically, several biologics, such as secukinumab, ixekizumab, bimekizumab, and brodalumab, target IL-17 to treat psoriasis [158]. Therefore, we propose using EVs as a novel therapeutic strategy for psoriasis by suppressing IL-17 and its downstream signaling pathways. Given its pivotal role in orchestrating immune responses and chronic inflammation in psoriasis, the NF-κB signaling axis has emerged as an indirect therapeutic target through inhibition of upstream cytokine pathways [159]. Various IL-17 inhibitors, such as secukinumab, ixekizumab, bimekizumab, and brodalumab, have been explored for their potential to treat psoriasis by blocking IL-17-mediated NF-κB activation, demonstrating encouraging results in alleviating cutaneous inflammation [160]. These agents function by blocking IL-17 signaling, thereby inhibiting downstream NF-κB activation and the transcription of pro-inflammatory genes [161].

First-generation IL-17 inhibitors target distinct components of the IL-17 pathway, yielding distinct immunosuppressive effects. While this may enhance efficacy in other patient populations, it also creates variations in safety profiles due to different mechanisms of action. Secukinumab, a fully humanized immunoglobulin G1 kappa (IgG1κ) monoclonal antibody, selectively neutralizes IL-17A by directly binding to this cytokine and preventing its interaction with IL-17 receptors [162]. By blocking IL-17A, secukinumab interrupts the downstream signaling cascade that involves activation of NF-κB through the recruitment of adaptor proteins such as Act1 adaptor protein (ACT1) and TNF receptor-associated factor 6 (TRAF6) [163]. Ixekizumab employs a similar mechanism, functioning as a humanized monoclonal antibody that targets IL-17A explicitly, thereby preventing the cytokine-induced activation of NF-κB and subsequent inflammatory gene transcription [164].

The molecular mechanism by which these IL-17 inhibitors indirectly suppress NF-κB involves interruption of the IL-17 signaling cascade. Upon IL-17 binding to its receptors, a complex signaling network is activated involving ACT1, TRAF6, and caspase recruitment domain-containing protein 14 (CARD14), which ultimately leads to IKK activation and subsequent NF-κB nuclear translocation [165]. By preventing IL-17 from engaging its receptors or by blocking the receptors themselves, these inhibitors effectively disrupt this inflammatory cascade, resulting in reduced NF-κB-dependent expression of pro-inflammatory cytokines, chemokines, and antimicrobial peptides that characterize psoriatic lesions [163].

IL-17 inhibitors exert their therapeutic activity in psoriasis predominantly by interrupting IL-17-driven activation of the NF-κB cascade, thereby dampening the pro-inflammatory cytokine networks that sustain KCs hyperproliferation and chronic cutaneous inflammation. In translational and clinical studies, blockade of IL-17 signaling has been shown to attenuate NF-κB-dependent transcriptional programs in epidermal and immune compartments, restoring terminal differentiation pathways and facilitating plaque resolution in vivo. Consequently, targeting IL-17 to modulate NF-κB signaling constitutes a rational, disease-relevant strategy that addresses both the immunologic drivers and epithelial dysfunction characteristic of psoriatic pathology.

### 5.3. IL-23/IL-17 Axis–Targeted Therapies

Biologic therapies for psoriasis, including TNF-α inhibitors, IL-17 inhibitors, and IL-23 inhibitors, are antibody-based treatments that have demonstrated substantial therapeutic efficacy. However, these biologics are associated with various adverse effects. A meta-analysis focusing on IL-17 and IL-23 inhibitors reported infections as the most common adverse event, occurring in approximately 33.16% of patients [166].

Psoriasis was traditionally viewed as a Th1-mediated disorder, largely due to elevated IFN-γ production by CD^4+^ T cells within psoriatic lesions and minimal expression of Th2-associated cytokines such as IL-4, IL-5, and IL-13 (Figure 1) [66]. However, the discovery of the Th17 cell subset was first identified in a mouse model of experimental autoimmune encephalomyelitis, commonly used to study multiple sclerosis [167], which prompted a significant shift in the dermatological understanding of chronic skin inflammation.

Th17 cells, distinguished by their secretion of IL-17 and IL-22, have since been recognized as central players in psoriasis pathogenesis (Figure 2) [66]. Notably, therapeutic inhibition of IL-17 has been shown to reverse both the molecular signatures and clinical symptoms of psoriasis in most patients, firmly positioning IL-17 and IL-17–producing T cells as core components of the current disease model. It is also important to note that in humans, IL-22 is predominantly produced by a distinct TH22 subset.

Multiple skin-resident immune cells, including Th17 (CD^4+^), cytotoxic T cell 17 (T_C_17) (CD^8+^), γδ T cells, and innate lymphoid cells, which produce IL-17, primarily in response to IL-23 secreted by dDCs [168,169,170]. Upon exposure to psoriasis-related autoantigens or environmental triggers (e.g., infection or trauma), these IL-17-producing cells release pro-inflammatory cytokines such as IL-17A/F, TNF, IL-26, and IL-29 [171]. These signals activate transcription factors, such as C/EBPβ/δ, STAT1, and NF-κB in KCs, initiating a self-amplifying inflammatory cascade. This ultimately leads to KCs hyperproliferation, leukocyte recruitment, and the formation of psoriatic plaques [66]. Moreover, IL-17 and TNF act synergistically to enhance pro-inflammatory gene expression (e.g., IL-1β, IL-6, IL-8), further promoting DCs activation and T17 cell differentiation in both skin and draining lymph nodes.

## 6. Extracellular Vesicles and Psoriasis Therapy

### 6.1. Stem Cell-EV in Psoriasis Therapy

Several studies have demonstrated that EVs derived from various types of stem cells, including UC-MSCs, embryonic stem cell-derived mesenchymal stem cells (ESC-MSCs), and adipose-derived mesenchymal stem cells (AD-MSCs), can help reduce psoriasiform skin inflammation [172].

Studies by Zhang and Lai et al. demonstrated the therapeutic potential of ESC-MSC-derived EVs in an IMQ-induced psoriasis mouse model [33,173]. Their findings revealed that these EVs primarily accumulated in the stratum corneum, unable to penetrate deeper into the epidermis due to the tight junctions within the stratum granulosum. Zhang et al. assessed a topical MSCs EVs cream as an anti-inflammatory therapy for psoriasis [173]. EVs (50–200 nm; hydrodynamic radius 55–65 nm) derived from an hESC-MSCs line contained canonical exosomal markers (CD81/CD9/Alix) and the complement inhibitor CD59 and were formulated in an oil-in-water emulsion [173]. In human skin explants, Alexa-488–labeled vesicles remained primarily within the stratum corneum with <1% fluorescence detected in the medium over 24 h, indicating minimal dermal penetration. In imiquimod (IMQ) mouse models, efficacy depended on disease burden and treatment duration: in a milder regimen (IMQ days 0–2; EVs days 3–9), the cream significantly reduced IL-17 and C5b-9 in skin versus vehicle, with a non-significant trend for IL-23; external phenotype scores were not different from spontaneous recovery. No benefit was observed in a severe regimen (IMQ days 0–5; EVs days 3–5) [174]. This study proposed a local mechanism in which EVs confined to the stratum corneum inhibit complement activation via exosomal CD59, thereby dampening C5b-9 formation, NETosis, and neutrophil-derived IL-17 (Figure 3). These findings suggest that topical MSCs EVs can attenuate key inflammatory mediators of psoriasis while potentially avoiding systemic exposure associated with oral or intravenous therapies.

A research work evaluated whether human UC-MSC-EVs mitigate psoriasis by suppressing the IL-23/IL-17 axis [175]. Purified 50–200 nm vesicles (CD9/CD63/CD81^+^) were administered subcutaneously (50 µg, days 0/2/4) in imiquimod-induced psoriatic C57BL/6 mice. UC-MSCs-EVs significantly improved disease severity, as indicated by lower PASI scores, reduced erythema, scaling, and thickening, and diminished epidermal hyperplasia and mononuclear infiltration [175]. Skin extracts from treated mice showed coordinated decreases in STAT3/p-STAT3, IL-17, IL-23, and CCL20. Mechanistically, UC-MSCs-EVs acted on two key cell types: (i) in bone-marrow–derived dendritic cells, maturation/activation was blunted (↓CD11c^+^MHCII^+^, ↓CD11c^+^CD86^+^) with suppressed IL-23p40 secretion; and (ii) in IL-17A–stimulated keratinocytes (HaCaT), STAT3 activation and downstream IL-17, IL-23, CCL20 output were reduced, disrupting the IL-17–driven positive feedback loop [175]. Collectively, the findings indicate that UC-MSCs-EVs attenuate psoriasis-like inflammation through convergent inhibition of DC-derived IL-23 and keratinocyte STAT3 signaling, supporting their development as a cell-free immunomodulatory therapy for chronic inflammatory skin disease.

A research team demonstrated a dual role for EVs in psoriasis [176]. Patient serum–derived EVs (PS-EVs; 123–148 nm, CD63/CD9^+^) were readily internalized by HaCaT keratinocytes and amplified inflammation and oxidative stress, upregulating IL-1β, IL-6, TNF-α, NOX2/NOX4, and downregulating Nrf2, via p65/NF-κB and p38/MAPK activation; Patient serum-derived EVs also impaired autophagy (↓ATG5, P62, Beclin1, LC3B, and fewer LC3B puncta) [176]. In contrast, AD-MSCs-EVs reversed these pathogenic effects, suppressing cytokines and NOX enzymes, restoring Nrf2, and reactivating autophagy (↑ATG5/P62/Beclin1/LC3B and LC3B puncta). Mechanistically, AD-MSCs-EVs attenuated PS-EVs–driven signaling through NF-κB/MAPK and rescued the autophagy–redox axis [176]. Together, the data identify circulating Patient serum-derived EVs as active drivers of keratinocyte dysfunction and highlight AD-MSCs-EVs as a cell-free, autophagy-modulating candidate therapy for psoriasis.

This research evaluated IFN-γ–stimulated UC-MSCs-EVs (IFNγ-sEVs) as both immunosuppressants and natural nanocarriers for an anti–miR-210 antisense oligonucleotide (ASO-210) in psoriasis [177]. Purified sEVs (CD63/CD9/TSG101/Alix^+^) were electroporated with-210 to generate ASO-210@IFNγ-sEVs, which were efficiently taken up by T cells and keratinocytes. Their study showed that ASO-210-loaded UC-MSCs-EVs exhibited an average particle size of 115 nm and a zeta potential of −22.48 mV. The research team demonstrated that UC-MSCs-EVs achieved a drug loading efficiency of 13.35% and released approximately 94.02% of the therapeutic payload within 48 h [177]. IFNγ-sEVs alone suppressed proliferation/activation of human PBMC and CD3^+^ T cells, accumulated in IMQ–injured skin, and ameliorated disease in IMQ mice (lower PASI, epidermal hyperplasia, Ki67, and inflammatory infiltrates), while reducing IL-17A/IFN-γ/IL-6/TNF-α and shifting splenic T cells from Th17 to Th2. Loading with ASO-210 potentiated these effects: ASO-210@IFNγ-sEVs produced the most significant clinical and histologic improvement, knocked down miR-210, upregulated STAT6 and LYN, and outperformed a simple ASO + sEV mixture, consistent with enhanced cargo protection, delivery, and lesion targeting [177]. Intradermal administration was well tolerated without systemic toxicity. Collectively, IFNγ-sEVs, and particularly ASO-210@IFNγ-sEVs, emerge as a cell-free, immune-modulating nanoplatform that corrects Th17-driven inflammation and holds promise for nucleic-acid-based therapy in chronic inflammatory skin disease.

Engineered EVs have emerged as promising delivery vehicles for psoriasis therapy due to their potential for targeted action. One notable strategy involves exploiting the PD-1/PD-L1 immune checkpoint pathway, which plays a critical role in downregulating immune responses and maintaining immunological balance [178]. The interaction between PD-L1 and PD-1 on T cells suppresses T-cell proliferation and reduces the secretion of proinflammatory cytokines. In this regard, a research team utilized lentiviral transduction to create PD-L1–overexpressing MSCs and harvested their EVs for therapeutic application in psoriasis [179]. These PD-L1^+^ MSC-derived EVs exhibited a uniform spherical shape with an average size of approximately 100 nm. In an IMQ-induced psoriasis mouse model, treatment with PD-L1-expressing EVs led to greater improvement in skin inflammation and lesion severity than EVs lacking PD-L1. The therapeutic benefits were linked to the restoration of Th17/Treg immune balance and suppression of key pro-inflammatory cytokines, including IL-17A, IL-6, IFN-γ, TNF-α, and IL-1β [179]. These findings support the potential of PD-L1-enriched MSCs EVs as a targeted and effective approach for managing psoriasis.

A research team has developed a multifunctional fused nanovesicle system to target inflamed skin lesions for the treatment of autoimmune skin disorders, such as psoriasis [180]. Their approach initiated with the encapsulation of CX5461, a small-molecule immunosuppressant known for its anti-proliferative effects, into GEVs using electroporation. To enhance therapeutic efficacy and safety, these GEVs were fused with CCR6-positive nanovesicles derived from the membranes of genetically modified gingival mesenchymal stem cells, yielding FV@CX5461 [180]. These hybrid vesicles exhibited a spherical morphology with intact membranes, an average size of approximately 160 nm, and a zeta potential of approximately -21 mV. The drug-loading capacity was reported to be around 18%, with approximately 90% of the payload released in vitro within 48 h. FV@CX5461 significantly downregulated pro-inflammatory cytokine production in HaCaT cells, suppressed Th17 cell activation, and promoted Treg cell infiltration [180]. Notably, the CCR6 on the surface of FV@CX5461 facilitated targeted delivery to CCL20-rich inflamed skin areas, thereby restoring immune balance and mitigating IMQ-induced psoriasis symptoms.

A recent study performed by Mohseni Meybodi et al. assessed the safety and efficacy of AD-MSCs-EVs in treating mild to moderate plaque psoriasis [181]. In this clinical trial, twelve patients were divided into three groups and received a single dose of EVs (50, 100, or 200 μg). The researchers did not observe significant adverse effects, except for mild discomfort in two patients [181]. Notably, skin thickness was reduced in the 100 and 200 μg groups, with erythema and induration significantly improved only at 200 μg. Immunohistochemical analysis showed decreased IL-17 and CD3 expression and increased Forkhead box P3 (FOXP3) expression in patients receiving 100 and 200 μg. Additionally, 200 μg of EVs downregulated IL-23 and TNF-α while upregulating IL-10 [181]. These findings suggest that a 200 μg dose is both safe and effective in early-phase treatment, improving clinical symptoms and modulating inflammatory responses. However, larger trials and exploration of higher or repeated dosing are needed to validate these outcomes.

### 6.2. Non-Stem Cell-Extracellular Vesicles in Psoriasis Therapy

Besides stem cell-derived EVs, those from other cellular sources have also demonstrated therapeutic potential against psoriasis. A research team investigated EVs derived from umbilical cord blood mononuclear cells (UCB-MNC-EVs) and found that they promoted the polarization of macrophages toward an anti-inflammatory state [182]. UCB-MNC-EVs also suppressed the proliferation of CD^4+^ and CD^8+^T cells and reduced the production of proinflammatory cytokines by peripheral blood mononuclear cells (PBMCs). Furthermore, UCB-MNC-EVs increased FOXP3 expression in PBMCs, suggesting a potential role in restoring the Th17/Treg immune balance. In an IMQ-induced psoriatic mouse model, they also alleviated acanthosis and decreased IL-17A and CCL20 expression, suggesting their potential as a therapeutic approach for Th17-mediated inflammatory skin disorders [182].

KCs, the predominant cell type in the epidermis, serve as a crucial barrier against external pathogens [183]. In psoriatic skin, the terminal differentiation of KCs is impaired, and the proliferation of KCs stem cells is abnormally regulated, leading to enhanced activation and expansion of KCs. KC-derived EVs carry bioactive molecules, including chemokines, miRNAs, and MHC proteins, with their content varying depending on the state and stimuli of the originating cells [184,185]. These KC-derived EVs are strongly implicated in the development of psoriasis. For example, when EVs from IL-17-stimulated KCs were applied to normal KCs, they triggered the upregulation of psoriasis-associated genes [186]. Moreover, EVs from KCs exposed to environmental pollutants were shown to activate the aryl hydrocarbon receptor pathway in HaCaT cells, leading to increased production of pro-inflammatory cytokines and chemokines [187].

A research team utilized a probe sonication technique to encapsulate TFC within KC-derived EVs (TFC-EVs) and evaluated their anti-psoriatic potential [188]. Among various preparation methods, the probe ultrasound approach yielded the highest drug loading efficiency (30.70%), with a zeta potential of −8.7 mV and a drug release rate of 67.5% over 24 h. Notably, TFC-EVs showed greater efficacy in reducing the expression of proinflammatory cytokines, including TNF-α, IL-23, IL-6, and IL-15, compared to free TFC [188]. In an IMQ-induced psoriasis mouse model, TFC-EVs also demonstrated superior therapeutic outcomes. These results highlight the potential of EVs as effective delivery vehicles to enhance the therapeutic impact of TFC in psoriasis treatment.

More recently, another study highlighted the role of patient serum-derived EVs carrying miR-6785-5p in alleviating psoriasis-like inflammation both in vitro and in vivo. These EVs were efficiently internalized by KCs and mitigated their hyperproliferative and inflammatory behavior by modulating the MNK2/p-eIF4E signaling pathway, ultimately reducing psoriatic symptoms.

KC-derived EVs also play a crucial role in intercellular communication, influencing various cellular functions involved in psoriatic skin inflammation. For instance, miRNAs carried within EVs from psoriatic KCs have been shown to promote T cell polarization, favoring differentiation into Th1 and Th17 subsets, which are key drivers of psoriasis pathogenesis [89]. These EVs also modulate DC phenotypes and cytokine profiles [189]. Research work demonstrated that KC-derived EVs activate both NF-κB and p38 MAPK signaling pathways, triggering the formation of NETs and stimulating neutrophils to release pro-inflammatory cytokines such as IL-6, IL-8, and TNF-α [57]. This response was found to exacerbate psoriatic symptoms in IMQ-induced mouse models.

In addition, KC-derived EVs modulate the behavior of dermal fibroblasts by regulating Extracellular matrix (ECM)-related proteins and activating pathways involved in fibroblast migration [190,191]. These EVs also increased fibroblast growth factor 2 expression in a concentration-dependent manner. Macrophage activation appears to be modulated through communication with KCs as well.

A study demonstrated that EVs enriched with leucine-rich α-2-glycoprotein 1, derived from the epidermis of psoriatic mice, activated macrophages through a TGF-β receptor 1-mediated mechanism, thereby aggravating psoriasiform skin inflammation [192]. Another study identified elevated levels of miR-4505 in the skin of individuals with psoriasis [193]. EVs derived from vitamin D receptor-deficient keratinocyte (EVs-shVDR), which contain miR-4505, were found to enhance macrophage proliferation and promote classically activated (M1) macrophage polarization while inhibiting apoptosis. These findings suggest that miR-4505 plays a key role in mediating the immunomodulatory effects of EVs-shVDR on macrophages.

Combining EVs with cell membranes or nanovesicles has emerged as a novel therapeutic strategy for treating psoriasis. This hybrid approach preserves the biological activity of drug-carrying EVs, although incorporating the functional characteristics of the fused membranes. Wang et al. developed fused EVs by integrating annexin A1-overexpressing T cell-derived EVs with membranes from M2 macrophages [194]. These engineered vesicles retained both the anti-inflammatory traits of M2 macrophages and the immunomodulatory effects of ANXA1. The resulting hybrid EVs exhibited a particle size of approximately 169 nm and a zeta potential of –11 mV. In vitro studies demonstrated efficient uptake of these vesicles by macrophages, leading to enhanced M2 polarization [194]. Furthermore, subcutaneous administration of the fused EVs in a psoriasis-like mouse model elevated expression of the M2 macrophage marker Arg1 and significantly reduced levels of inflammatory cytokines, including IL-1β, IL-6, and TNF-α, in affected skin [194]. These findings suggest that membrane-fused EVs represent a promising approach for modulating immune responses and alleviating psoriatic inflammation.

Figure 4 illustrates the therapeutic potential of mesenchymal stromal/stem cell–derived extracellular EVs in psoriasis. The top panel highlights the diversity of MSCs sources, including umbilical cord, adipose tissue, and other origins, that can be harnessed to generate a standardized, cell-free product: MSCs-EVs. These vesicles, enriched in bioactive cargo such as miRNAs, proteins, and lipids, represent a scalable therapeutic platform that retains MSCs immunomodulatory properties while avoiding the safety and handling concerns associated with live-cell therapies. The middle panel contrasts normal versus psoriatic epidermis, emphasizing the pathological hallmarks of psoriasis, including epidermal hyperplasia, elongated rete ridges, leukocyte infiltration, and excessive NF-κB activity in KCs, driving pro-inflammatory cytokine release (e.g., TNF-α/IL-17). The bottom panel depicts the mechanism of action of MSCs-EVs following delivery via topical, intradermal, or systemic routes. EVs traffic to lesional skin, where KCs and immune cells take them up, suppress NF-κB signaling, reduce inflammatory mediators, and restore KCs proliferation and differentiation toward a more physiological balance. Collectively, this figure conveys the central message that MSCs-EVs provide a standardized, cell-free, and multipotent therapeutic approach that can attenuate psoriatic inflammation and promote tissue repair, thereby offering strong translational potential for inflammatory skin diseases.

### 6.3. Engineered Extracellular Vesicles-Based Psoriasis Therapy

These nanosized EVs, typically 30-150 nanometers in diameter, are encapsulated in a lipid bilayer and secreted by numerous cell types [195]. EVs express unique surface markers such as CD81, CD63, and CD9, and they transport a diverse array of bioactive molecules, including DNA, mRNA, microRNAs (miRNAs), proteins, lipids, and metabolites [196]. These molecular cargos play significant roles in modulating metabolic processes, cellular proliferation and differentiation, immune responses, and intracellular signaling pathways. Effective intercellular communication is essential for the onset and progression of various diseases. In recent years, EVs have emerged as critical facilitators of cell-to-cell communication [197].

In psoriasis, EVs derived from both immune and non-immune cells within affected skin lesions are thought to be actively implicated in the disease development [198]. Their contents can alter gene expression and activate signaling pathways in recipient cells, thereby accelerating the progression of psoriatic pathology [187,192,199]. EV-based psoriasis therapy has garnered significant interest for its potential therapeutic use in autoimmune conditions such as psoriasis, primarily because of its ability to modulate immune responses [200].

Recently, antibody-based therapies have demonstrated efficacy in treating psoriasis. However, these therapies are associated with several limitations, including high costs, systemic side effects, and reduced drug survival rates [201,202]. Consequently, there is a growing need for safer and more precise therapeutic strategies. In this context, EVs, nano-sized vesicular structures secreted by various cell types, have attracted attention for their ability to mediate intercellular communication by carrying biological molecules [203].

Remarkably, EVs have shown potential not only as diagnostic biomarkers but also as therapeutic agents, given their capacity to suppress pro-inflammatory macrophages and influence T-cell differentiation [204,205]. However, the direct use of natural EVs as therapeutics faces challenges, such as limited target specificity and suboptimal therapeutic efficacy [206,207]. To overcome these limitations, recent efforts have focused on engineering EVs through various bioengineering techniques to enhance their therapeutic potential.

To enhance the therapeutic efficacy of EVs, strategies for gene and drug loading are being actively explored. For example, EVs derived from AD-MSCs, loaded with NF-κB siRNA (si-AD-MSCs-EVs), promoted the polarization of macrophages toward the M2 phenotype by increasing CD206 expression (an M2 marker) [208]. In contrast, unmodified EVs did not induce significant changes. The si-AD-MSCs-EVs also exhibited twice the inhibitory capacity against pro-inflammatory cytokines compared to unmodified EVs and, in an Ultraviolet B (UVB)-induced skin lesion model, reduced epidermal thickness and attenuated inflammatory cell infiltration [208].

Engineered EVs can also enhance the therapeutic efficacy of conventional drugs. In one notable study, EVs derived from KCs were loaded with the JAK inhibitor TFC using probe sonication. Compared to free TFC, these engineered EVs demonstrated 4 times greater safety, reducing cytotoxicity, and exhibited twice the inhibitory effect on pro-inflammatory gene expression. In the IMQ-induced psoriasis model, epidermal thickness was reduced by 46%, confirming that low-dose TFC delivered via EVs achieved comparable therapeutic effects with reduced toxicity [188].

Innovative strategies utilizing hybrid EV systems have also been reported. In one example, grapefruit-derived EVs -like nanovesicles (GEVs) were loaded with CX5461 via electroporation and fused with CCR6^+^ gingiva-derived MSCs membranes by extrusion to produce engineered EVs. These EVs suppressed Th17 responses and promoted Treg differentiation by blocking the JAK-STAT/T-cell receptor (TCR) and potentially the Phosphatidylinositol 3-kinase (PI3K)/Protein kinase B (AKT) and MAPK signaling pathway. In the IMQ-induced psoriasis mouse model, this approach reduced epidermal thickness, inhibited infiltration of CD3^+^ T cells and CD4^+^ T cells, lowered serum IL-17A and IFN-γ levels, and normalized spleen indices, demonstrating systemic anti-inflammatory effects [180].

Another major challenge for EV-based therapies is achieving precise targeting of inflamed tissues where therapeutic action is required. To address this, technologies have been developed to attach specific ligands or antibodies to the EV surface or to fuse EVs with particular cell membranes [11]. For example, EVs engineered to express programmed death-ligand 1 (PD-L1) on their surface can selectively bind to programmed death protein 1 (PD-1) receptors at sites of inflammation, thereby modulating T cell responses [12,13]. In addition, T cell-derived EVs overexpressing the anti-inflammatory protein Annexin A1 (ANXA1) and fused with M2 macrophage membranes demonstrated significantly greater tissue accumulation and immunoregulatory effects compared to natural EVs [14].

These modifications not only enhance target specificity but also improve therapeutic efficacy. In an IMQ-induced psoriasis mouse model, melanoma cell-derived EVs loaded with pristimerin and engineered to express PD-L1 effectively suppressed Th17 cells, increased Treg cells, and significantly reduced epidermal thickening and T-cell infiltration. This approach showed superior PD-L1 expression levels and T cell inhibition compared to natural EVs [209]. Similarly, T cell-derived EVs overexpressing ANXA1 demonstrated improved psoriasis suppression in the IMQ-induced model, as evidenced by reduced skin lesions, inhibited pro-inflammatory cytokine secretion, and alleviation of liver and kidney damage, outperforming conventional EVs [194].

Psoriasis is not a disease confined to the hyperactivation of a single cytokine or cellular pathway; instead, its pathogenesis involves complex interactions among various immune cells, including Th1/Th17 cells, DCs, macrophages, and KCs [210]. Consequently, multi-pathway inhibition strategies utilizing EVs have garnered significant attention. For example, a study using electroporation to load the arginase-1 (Arg1) inhibitor nor-NOHA into MSC-derived EVs successfully suppressed both the Arg1/polyamine pathway and NF-κB signaling. This dual inhibition regulated the inflammatory responses of KCs and DCs, ultimately reducing Th17 cell infiltration [211]. Similarly, EVs derived from UVB-exposed cells and loaded with the mechanistic target of rapamycin (mTOR) inhibitor JPH203 concurrently blocked leucine uptake to suppress UVB-induced mTOR signaling and inhibited NF-κB via interleukin-1 receptor A. This approach promoted the restoration of the immune-skin axis by normalizing KCs differentiation and increasing IL1RA expression [212].

A key clinical limitation of psoriasis is the lack of reliable biomarkers for diagnosis and assessment of therapeutic response. However, EV-based biomarkers offer promising potential to address this gap [205]. Engineered EVs are emerging as next-generation immunotherapy platforms capable of overcoming the shortcomings of existing treatments by enabling targeted delivery, multi-pathway inhibition, and gene-based immune modulation [213]. Various strategies have demonstrated superior target specificity, loading efficiency, biostability, and therapeutic efficacy compared to natural EVs, with some approaches achieving the safety and reproducibility levels necessary to advance toward clinical trials [214,215]. Future challenges include establishing standardized large-scale production methods, conducting pharmacokinetic studies, and verifying long-term safety. Notably, the therapeutic potential of engineered EVs extends beyond psoriasis to a broad range of immune-mediated diseases, including atopic dermatitis, lupus, and rheumatoid arthritis [216,217].

Natural EVs have demonstrated anti-inflammatory potential by modulating key pathways, such as NF-κB and JAK/STAT, in multiple studies. However, their limited targeting efficiency and capacity to modulate only a single inflammatory pathway have constrained their clinical translation [188]. Recent advances in EV engineering, including drug loading, surface functionalization, and hybrid system integration, aim to overcome these limitations and enable the simultaneous targeting of multiple inflammatory pathways.

In a Phase I clinical trial, topical application of PTD2021P EV ointment to the forearm for 20 days in healthy volunteers was well tolerated, with no significant adverse events. Pre-clinical studies have shown that MSCs’ EVs can inhibit the C5b-9 complex, reducing pro-inflammatory molecules such as IL-17, suggesting a potential therapeutic mechanism for psoriasis [218]. Small EVs derived from MSCs overexpressing PD-L1 reduced epidermal thickness by about 55.5% in a psoriasis animal model. This effect was attributed to a dual mechanism involving PD-1/PD-L1 immune checkpoint modulation and suppression of both NF-κB and JAK/STAT signaling pathways [179]. Table 1 summarizes the role of EVs in psoriasis treatment and their possible mechanisms.

## 7. Challenges and Future Perspectives

There is no single global standard for grading psoriasis severity; guidelines use combinations of Body Surface Area (BSA), Psoriasis Area Severity Index (PASI), Dermatology Life Quality Index (DLQI), Physician Global Assessment, and special-site involvement and generally dichotomize disease as mild versus moderate-to-severe. The National Psoriasis Foundation stratifies patients by BSA to guide topical versus systemic therapy [219], the American Academy of Dermatology adds special-area involvement to its mild/moderate/severe categories [220], the European Consensus Program integrates BSA, PASI, and DLQI [221], and the International Psoriasis Council emphasizes BSA and special sites [222].

Treatment is severity-dependent. For mild disease, topicals (corticosteroids, vitamin D analogs, calcineurin inhibitors, tapinarof cream) are first line [223]. Moderate-to-severe psoriasis typically requires phototherapy, systemic agents, or biologics; narrowband UVB is preferred over PUVA for safety/efficacy [224]. Oral agents (methotrexate, cyclosporine, acitretin, apremilast, deucravacitinib) remain widely used for cost and convenience [225]. Biologics targeting IL-17/IL-23 achieve superior efficacy versus conventional drugs but face constraints of cost, adverse effects, heterogeneous response, and waning durability [174]. These limitations highlight the need for novel therapeutic approaches.

EVs can rationally complement biologics and small molecules by expanding pathway coverage, pairing IL-23/IL-17 blockades with upstream NF-κB/JAK inhibition, and by enabling dose reduction to mitigate class-specific adverse events. Local delivery (topical or intradermal) of EVs co-formulated with JAK or p38 inhibitors may yield additive anti-inflammatory effects while minimizing systemic exposure. Engineered EVs loaded with anti-inflammatory miRNAs or anti-miR oligonucleotides can precisely modulate dendritic-cell IL-23 production and keratinocyte STAT3 activity; clinically, they could be used as an add-on therapy to establish biologics to allow dose reduction while maintaining efficacy.

To mitigate translational risk, we propose a structured safety plan that addresses the full risk spectrum in a transparent, reproducible manner. Unintended immunosuppression is minimized by favoring local (topical/intradermal) delivery, step-up dosing, latent-infection screening, and predefined stopping rules. Off-target biodistribution is controlled through tissue-targeting ligands and quantitative biodistribution with immunophenotyping to map cellular uptake.

Pro-inflammatory or pro-coagulant contaminants are mitigated via orthogonal purification (e.g., ultrafiltration), endotoxin/pyrogen and tissue-factor testing, and confirmed sterility/mycoplasma clearance. Allo-immunogenicity and repeat-dose tolerance are managed by source standardization, HLA profiling, anti-EV antibody monitoring, and cytokine-release testing. Chemistry, manufacturing, and controls (CMC) consistency is ensured by releasing products against MISEV-aligned identity/purity panels (CD9/CD63/CD81, TSG101/ALIX, negative markers), verifying particle-to-protein ratios and mode diameter, performing cargo fingerprinting, and validating potency assays.

Combination trials should begin with an EV-alone lead-in, then progress to add-on with reduced-dose biologic or small molecule, incorporating drug–EV interaction assessments, dermatology-relevant endpoints (PASI, DLQI), and prespecified infection/pharmacovigilance monitoring.

Stem cell-derived EVs have the potential to open new horizons in psoriasis treatment due to their potent immunomodulatory and regenerative properties. However, successfully developing and applying them as clinical therapeutics requires overcoming key technical challenges, namely large-scale EV production, high-purity isolation and purification, and the standardization and quality control of the final product.

Appropriate guidelines for the therapeutic dose of EVs in clinical trials and commercialization are still lacking, and dosing standards are inconsistent, leading to preclinical and clinical trials using various metrics, such as EV particle count, protein content, and cell equivalence. A system capable of reliably mass-producing the high concentration of EVs required for patient administration is essential.

Ongoing clinical trials typically administer stem cell-derived EVs at concentrations of ~70–200 μg/mL [181,218]. However, culturing stem cells in a conventional two-dimensional (2D) culture environment faces the fundamental limitation of severely restricted EV production yield, which is insufficient for adequate clinical use. To address this large-scale production problem, innovative strategies to increase EV yield from the stem cell CM are being actively researched.

The main strategies for promoting EV secretion are divided into physical stimulation, which involves subjecting stem cells to physical environmental stress or incorporating systems suitable for mass culture, and chemical factors and environmental modulation, which include regulating specific medium characteristics or administering transient stimulation to the stem cells [226]. Physical approaches involve culturing cells on a 3D platform, which increases cell–cell and cell–matrix interactions, thereby enhancing the secretion rate of EVs [227]. These methods include the Hanging drop method, where cells are suspended and cultured upside down in a hanging drop on a culture plate [228], the microwell array, where cells are seeded into small wells, typically 100–500 μm in diameter, the hydrogel, which swells and cross-links with water to mimic ECM characteristics [229], and the Bioreactor, which precisely controls continuous flow, temperature, and oxygen in a suspension state [230].

Chemical and environmental modulation approaches increase EV secretion rates by directly regulating the cellular stress response, the activation of defensive mechanisms, and EV biogenesis pathways [231,232]. Methods used include exposure to hypoxia [233], increasing intracellular calcium [234], priming with specific chemical factors [235,236], and adding growth factors and cytokines [231,237]. Physical and chemical stimulation strategies, such as 3D culture, bioreactors, cytokines, and specific chemical factor preconditioning, significantly increase yield by two-fold to several tens of fold, making them an essential prerequisite for the clinical application and commercialization of therapeutics.

However, this optimization process raises new challenges. The combination of each culture method and stimulation condition affects not only the EV production yield but also the composition of the intrinsic cargos (nucleic acids and proteins), which determines therapeutic efficacy, thereby deepening batch-to-batch EV heterogeneity. Furthermore, there is a potential risk that final products may be contaminated with impurities exhibiting toxicity or tumorigenicity due to added chemical factors or Hypoxia-inducing mechanisms. For the successful clinical application of EVs, a protocol must be established to ensure EV yield, a combination that maximizes specific cargo loading, and functional consistency.

In some research areas, such as disease biomarker analysis, EVs may be studied directly in the source matrix without the need for separation or concentration [238,239]. However, for clinical applications such as psoriasis treatment, the purification and isolation process to secure a sufficient EV particle count and a high-purity EV formulation from the CM is the second critical challenge that must be addressed for successful clinical translation. Physical, solubility, and biochemical properties broadly categorize EV isolation and purification technologies. Ultracentrifugation is the most widely used method, separating particles of different sizes and densities based on sedimentation rate, but it is limited by the need for expensive equipment, long processing times, and potential contamination by protein and lipoprotein [240]. Density gradients (DGs) use gradient materials, such as sucrose, to create density layers that separate particles based on their Buoyant density. While offering the advantage of very high purity, they are limited by the labor required for preparing the media solution [241]. Size-exclusion chromatography (SEC) uses the principle that larger particles traverse fewer pores in the porous column matrix and thus travel a shorter path, eluting faster compared to smaller particles. Separation based on particle size offers advantages such as high purity, rapid processing time, and preservation of functional integrity due to separation primarily by gravity, but low yield is often reported [242]. Filter concentration (FC) utilizes a filter membrane with a molecular weight cut-off to separate and concentrate EVs based on size. Although it offers the advantage of relative simplicity and speed for isolating EVs from many samples, contamination by protein or potential EV damage is possible [243].

Precipitation (e.g., with polymers such as PEG) aggregates EVs for sedimentation. It boasts high yield and is user-friendly, but suffers from low purity, severe contamination, and is costly for commercial use [244]. Finally, Immuno-precipitation (IP) or Affinity-precipitation (AP) separates EVs via antigen–antibody specific recognition and binding to EV surface proteins. It offers high purity and specificity with no specialized equipment required, but is disadvantaged by low yield and the need for expensive antibodies [240]. Therefore, for the successful clinical application of EVs, it is essential to maximize EV yield and specific and potential cargo loading, in addition to establishing purification and isolation steps to extract uncompromised, pure EVs.

For commercialization, it is vital to extract high-purity EVs from large-volume samples in a short period of time. This goal will only be achievable through multi-stage hybrid strategies rather than single techniques. Specifically, utilizing Tangential Flow Filtration (TFF), a Filter concentration (FC) technology capable of handling several L of sample, for primary separation and concentration, followed by high-resolution techniques such as SEC to remove non-EV contaminants and increase purity, is considered a core strategy for successful clinical entry.

Clinical utilization of EVs remains uncommon, mainly due to intrinsic heterogeneity and the resulting challenges in standardization and quality control (e.g., batch-to-batch variability, incomplete analytical characterization, and undefined potency assays) [245]. Nevertheless, multiple preclinical studies have demonstrated robust efficacy signals, and a small number of recent clinical trials in psoriasis have begun to confirm safety and preliminary efficacy [181,218]. Moving toward translation will require rational EV purification, prioritizing cargos that modulate the IL-23/IL-17 axis alongside clear decisions on timing, route, and dosing (e.g., local vs. systemic administration; protein/particle-based dose metrics), as well as validated release criteria and clinical endpoints aligned with the mechanism of action.

Figure 5 depicts an overview of how EVs can interrupt the psoriatic feed-forward circuit at multiple control points. Clockwise, the schematic traces a canonical cascade—from upstream triggers and DC production of IL-23, through JAK2/TYK2–STAT3 signaling that expands Th17 cells and amplifies IL-17, to proximal TRAF6/IKK activation and nuclear NF-κB–driven inflammatory outputs. The red T-bars indicate nodes where EV cargo (e.g., anti-inflammatory miRNAs, proteins such as CD59) has been shown to suppress pathway activity, thereby weakening DC IL-23, keratinocyte STAT3, and NF-κB–dependent cytokine programs and ultimately reducing IL-17. Approved agents (e.g., anti-IL-17 biologics; JAK/TYK2 inhibitors) are placed alongside their pharmacologic targets to highlight concordance between druggable nodes and EV mechanisms, underscoring opportunities for dose-sparing or combinatorial strategies. Taken together, EVs do not act at a single step but modulate multiple nodes along the IL-23/IL-17– and NF-κB–associated axes that sustain psoriatic inflammation.

## 8. Conclusions

In psoriasis, MSC-derived EVs show the most consistent therapeutic signal and the broadest mechanisms, attenuating dendritic-cell IL-23 production, suppressing keratinocyte STAT3 signaling, and (topically) limiting complement/NETosis. In contrast, non-stem-cell EVs are more source-specific: immune-cell EVs chiefly reprogram leukocytes (M2/Treg bias), and keratinocyte EVs are often pathogenic unless re-engineered as drug carriers. Given their strength, versatility, and early clinical safety, MSC-EVs are the leading platform, with engineered variants (targeting ligands or nucleic acids) recommended for added potency and specificity. We suggest local delivery (topical/intradermal) where feasible and biologic-sparing combinations to reduce systemic exposure. Development should be anchored in standardized CMC/safety metrics (MISEV-aligned analytics, potency assays, infection monitoring, and biodistribution monitoring). Prospective comparative and combination trials to assess PASI/DLQI and cytokine endpoints are warranted to identify the optimal strategies and confirm long-term durability.

Stem cell-derived EVs are emerging as safe, immunomodulatory candidates for psoriasis, capable of attenuating IL-23/JAK–STAT and IL-17/NF-κB signaling and rebalancing the Treg/Th17 axis; they may function as a monotherapy or as adjuncts that enhance efficacy and reduce toxicity of existing regimens. Yet biologic therapies can precipitate paradoxical dermatoses (psoriasis/eczema/pustular/lichenoid), prompting discontinuation in ~46% of affected cases, and carry concerns about cardiovascular events and malignancy, underscoring the need for alternatives [246,247,248]. While MSC and CM approaches show anti-inflammatory promise, they are constrained by limited clinical evidence, high production costs, variable durability, and, for direct MSC transplantation, risks of tumorigenicity and immunogenicity, procedural complexity, and poor post-graft survival/rapid clearance [249,250,251]. CM-based strategies mitigate cell-related risks but still face uncertain active constituents, potential off-target dissemination, low in vivo stability, and short half-life [115,250,252,253].

Stem cell-derived EVs show the most consistent anti-psoriatic activity, acting on shared pathogenic nodes via defined cargos, for example, miR-146a targeting IRAK1/TRAF6 to restrain NF-κB, miR-124/let-7 attenuating STAT3 signaling, and surface CD59 inhibiting C5b-9 formation to reduce NETosis, collectively lowering IL-23/IL-17 outputs [254,255]. In contrast, keratinocyte EVs are typically pathogenic (propagating NF-κB/p38 activation and Th1/Th17 polarization) unless engineered as lesion-tropic drug carriers [204,256,257,258]. We distinguish generalization (convergence on dendritic-cell IL-23, keratinocyte STAT3, NF-κB, and topically, complement/NETosis) from specificity (source-biased cargos and targets) and provide a concise table/figure mapping EV source, exemplar cargo, molecular target, pathway node, cell type, and phenotypic readout to anchor these mechanistic comparisons.

Framing EVs, the bioactive core of CM, as therapeutics may bypass several of these barriers, provided that manufacturing hurdles are overcome. Key challenges include low per-cell vesicle output, which necessitates scalable bioprocessing, and source/cargo heterogeneity, which demands rigorous batch-to-batch comparability [259,260].

In response, optimized culture/bioreactor platforms and consensus quality frameworks (e.g., ISEV’s MISEV guidelines) are standardizing identity, purity, and potency metrics to enable reproducible translation [261]. Prospectively, engineered EVs—incorporating drug/oligonucleotide payloads and surface targeting—together with precision delivery (topical, intradermal, or systemic with controlled biodistribution) could enable patient-tailored interventions that preserve anti-psoriatic efficacy while minimizing systemic risk. With ongoing advances in scale-up manufacturing, quantitative analytics, and clinical validation, MSC-derived EVs are well-positioned to evolve into next-generation, cell-free therapies for psoriasis.

## Figures and Tables

**Figure 1 ijms-26-10297-f001:**
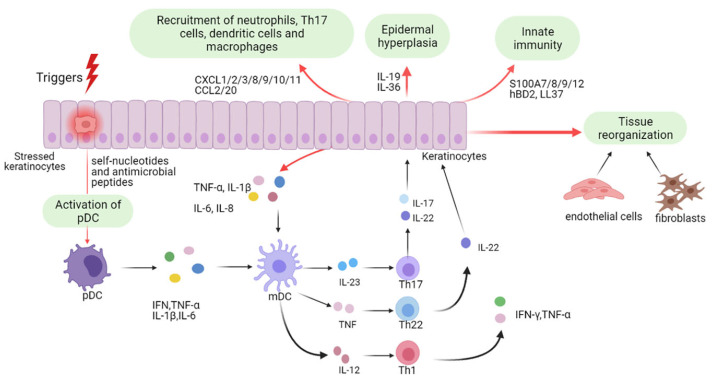
Keratinocyte-Mediated Pathogenic Mechanisms in Psoriasis. Schematic diagram illustrating psoriasis pathogenesis from the keratinocyte viewpoint. Triggered keratinocytes release nucleotides and antimicrobial peptides, activating plasmacytoid and subsequently myeloid dendritic cells, thus initiating disease. Upon cytokine stimulation, activated keratinocytes drive pathology through leukocyte infiltration, epidermal hyperplasia, innate immune activation, and tissue remodeling. Abbreviations: pDCs, plasmacytoid dendritic cells; mDCs, myeloid dendritic cells; IFN, interferon; TNF-α, tumor necrosis factor-α; IL-1β, interleukin-1β; Th1, T helper 1. This figure is reproduced from [68]. Copyright © 2022, The Author(s). This is an open access article distributed under the terms of the Creative Commons CC BY license.

**Figure 2 ijms-26-10297-f002:**
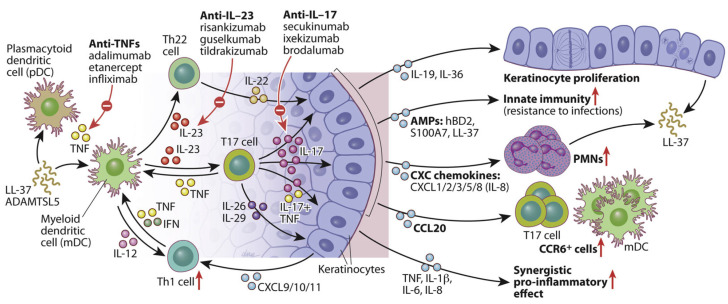
A schematic diagram illustrating the role of IL-23/T17 axis in modulating KCs activity in psoriatic skin. Elevated IL-23 and IL-17 signaling initiate a cascade of immune responses and alter KCs behavior. Under IL-23 regulation, IL-17 exerts multiple effects on epidermal KCs, including the indirect promotion of epidermal thickening via IL-19 and IL-36, and the enhancement of innate immune functions through upregulation of antimicrobial peptides such as hBD2, S100A7, and LL-37. IL-17 also facilitates the recruitment of immune cells, including neutrophils and mDCs, by stimulating the release of KC-derived chemokines. Additionally, it promotes the expression of key pro-inflammatory cytokines, such as IL-1β, IL-6, and IL-8, which act in concert with TNF to perpetuate the chronic inflammatory cycle characteristic of psoriasis. This figure is reproduced from [66] with permission. © 2017 American Academy of Allergy, Asthma & Immunology. Arrow up means upregulation.

**Figure 3 ijms-26-10297-f003:**
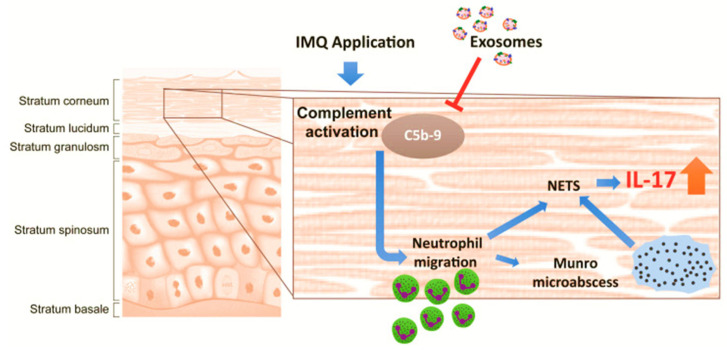
This diagram depicts how topically applied MSC-derived exosomes accumulate in the stratum corneum and suppress complement activation triggered by IMQ, thereby reducing NETosis and IL-17 release via NETs. The T arrow indicates the inhibitory effect on C5b-9 complex formation. This figure is reproduced from [173] with permission. This article is an open access article distributed under the terms and conditions of the Creative Commons Attribution (CC BY) license (http://creativecommons.org/licenses/by/4.0/). Blue arrow: Trace the cascade & Orange arrow indicates suppression.

**Figure 4 ijms-26-10297-f004:**
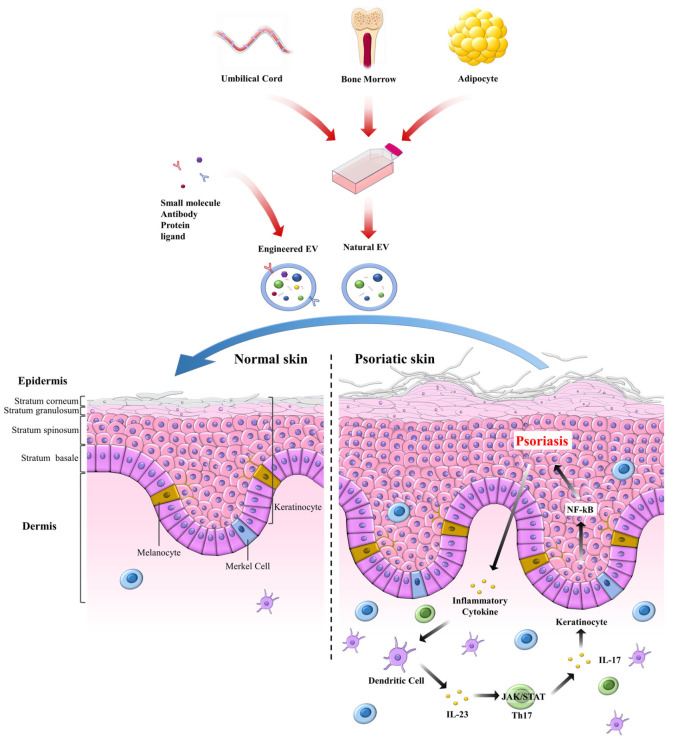
Schematic of a cell-free psoriasis therapy based on MSC-derived EVs. EVs are isolated from various MSC sources, such as umbilical cord (UC-MSCs), adipose tissue (AD-MSCs), etc., to generate a unified product. EVs are often engineered with small molecules, antibodies, proteins, or ligands to enhance therapeutic efficacy. In psoriatic lesions, hyperactive NF-κB signaling in KCs sustains inflammation and epidermal thickening. Administered MSCs-EVs home to lesional skin, deliver bioactive cargo, attenuate NF-κB activity, curb inflammatory cell crosstalk, and rebalance KCs growth and differentiation, thereby shifting tissue toward a healthy phenotype.

**Figure 5 ijms-26-10297-f005:**
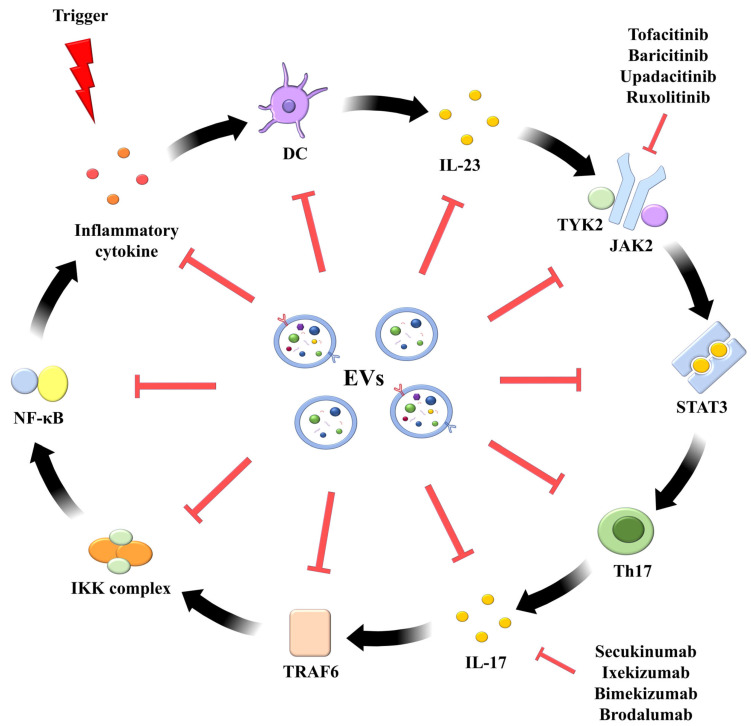
Extracellular vesicles exert broad suppressive effects on key pro-inflammatory mediators within the psoriatic positive feedback loop. Extracellular vesicles (EVs) derived from various stem cell sources exhibit potent inhibitory effects on multiple pathological factors implicated in the positive feedback loop of psoriasis pathogenesis. Specifically, EVs actively suppress central pro-inflammatory cytokines, such as IL-23 and IL-17, as well as crucial intracellular signaling molecules, including JAK2, STAT3, and NF-κB. Unlike conventional biologic agents that often target single pathways (e.g., specific JAK inhibitors or IL-17 blockers), EVs offer a pleiotropic therapeutic strategy by concurrently modulating a diverse array of inflammatory mediators. This multifaceted intervention restores immune homeostasis in affected tissues, thereby conferring comprehensive therapeutic benefit in psoriasis.

**Table 1 ijms-26-10297-t001:** Therapeutic potential of EVs for psoriasis.

Origin and Characteristics of EVs	Experimental Model	Main Activities and Mechanisms	Ref.
si-AD-MSCs-EVs -EV source cell: AD-MSCs -Cargo: NF-κB siRNA -Isolation method : differential ultracentrifugation -Size/Markers : 30 nm~200 nm (130 nm peak)/CD9, CD63, Alix	-Experimental model : RAW 264.7 cells (In vitro) -Route of administration: Incubation with cell culture medium -Dosage: -	-Inhibition of RAW 264.7 proliferation by 33.3% (*p* < 0.001) -Decrease in IL-6 by 45% (*p* < 0.001) -Decrease in TNF-α by 40% (*p* < 0.001) -Decrease in NF-κB mRNA by 55% (*p* < 0.001)	[208]
	-Experimental model : Epithelial cells (In vitro) -Route of administration : Incubation with cell culture medium -Dosage: -	-Decrease in NF-κB mRNA by 60% (*p* < 0.001)	
	-Experimental model: UVB-irradiation–induced skin lesion model in C57BL/6 mice (In vivo) -Route of administration: Topical application -Dosage: 200 μL (1 μg/mL), 3 times per week for 2 weeks.	-Decrease epidermal thickness (-) -Increase in CD206 (-) -Decrease in NF-κB (-) -Decrease in TNF-α by 60–70% (*p* < 0.001) -Decrease in IL-6 by 70–80% (*p* < 0.001) -Decrease in p-ERK (-) -Decrease in p-c-Jun (-)	
TFC-EVs -EV source cell: A-431 human epidermoid carcinoma cell line -Cargo: JAK inhibitor, TFC -Isolation method: EXOCIB isolation kit (polymer precipitation method) -Size/Markers: 55.2 ± 12 nm (hydrodynamic diameter)/CD9	-Experimental model: 5% IMQ induced psoriasis in BALB/c mice (In vivo) -Route of administration: Topical application -Dosage: TFC-EVs 2% in cold cream for 4 days (~0.0349 μg/day (ear), ~0.1046 μg/day (back) [EVs protein])	-Decrease in epidermal thickness by 46.1% (*p* < 0.001) -Decrease in PASI 48.7% (*p* < 0.001) -Decrease in Hyperkeratosis by 73.7% (*p* < 0.0001) -Decrease in Parakeratosis by 73.7% (*p* < 0.0001) -Decrease in Acanthosis by 73.7% (*p* < 0.0001) -Restoration in Tissue collagen by 75% (*p* < 0.0001) -Decrease in CD3 infiltration by 74.4% (*p* < 0.0001)	[188]
FV@CX5461 -EV source cell: Engineered gingiva-derived mesenchymal stem cells (GMSCs) fused with grapefruit-derived EV-like nanovesicles (GEVs) -Cargo: Encapsulated immunosuppressant CX5461, along with inherent anti-inflammatory/antioxidant lipids, natural metabolites, and miRNAs from GEVs, and CCR6 surface protein for targeting inflammatory sites -Isolation method: GEVs were first loaded with CX5461 via electroporation, then fused with CCR6-overexpressing GMSC-derived nanovesicles (prepared by membrane extrusion) through a 0.22 μm polycarbonate membrane extrusion -Size/Markers: 163.4 nm/Alix, TSG101, CD63, CD81	Experimental model : LPS-stimulated HaCaT cells (In vitro) -Route of administration : Incubation with cell culture medium -Dosage: 10 μg/mL	Decrease in ROS (-) Decrease in IL-6 by 72.7% (*p* < 0.001) Decrease in IL-1b by 71.4% (*p* < 0.001) Decrease in TNF-a by 70% (*p* < 0.001)	[180]
	-Experimental model: Human Peripheral Blood Mononuclear Cells (In vitro) -Route of administration: Incubation with cell culture medium -Dosage: 10 μg/mL	-Inhibition of PBMCs proliferation by 21.64% (*p* < 0.001)	
	Experimental model: RAW 264.7 cells (In vitro) -Route of administration: Incubation with cell culture medium -Dosage: 10 μg/mL	-Decrease in ROS (-) -Increase in CD206 by 1885% (*p* < 0.001)	
	-Experimental model: PMA/Ionomycin-stimulated Jurkat cells (In vitro) -Route of administration: Incubation with cell culture medium -Dosage: 10 μg/mL	-Decrease in p-JAK1 by 55.8% (-) -Decrease in p-JAK2 by 69.1 (-) -Decrease in p-STAT3 by 63.3% (-) -Decrease in p-LCK by 53.2% (-) -Decrease in p-AKT by 59.3% (-) -Decrease in p-ZAP70 by 56.1% (-) -Decrease in p-p38 MAPK by 60.2% (-)	
	-Experimental model: 5% IMQ induced psoriasis in BALB/c mice (In vivo) -Route of administration: IV administration via the tail vein -Dosage: 20 mg/kg GEVs protein equivalent + 2 mg/kg CX5461 for 3 days	-Decrease epidermal thickness by 82.35% (*p* < 0.001) -Decrease in IL-17A by 66.7% (*p* < 0.001) -Decrease IFN-γ by 75% (*p* < 0.001) -Decrease in CCL20 by 59.09% (*p* < 0.001) -Decrease in IL-1β by 64.29% (*p* < 0.001) -Decrease TNF-α by 65.52% (*p* < 0.001) -Decrease IL-22 by 66.67% (*p* < 0.001) -Decrease IL-12/23 p40 by 58.33% (*p* < 0.001) -Decrease in ROS by 75% (*p* < 0.001)	
Pri@EVs -EV source cell : mouse melanoma cell line B16-F10 -Cargo: pristimerin -Isolation method: differential ultracentrifugation -Size/Markers: 89.56 ± 26.35 nm/TSG101, CD63	-Experimental model: TNF-α stimulated HaCaT cells (In vitro) -Route of administration: Incubation with cell culture medium -Dosage: 20 μg/mL	-Inhibition of HaCaT proliferation by 33.3% (*p* < 0.0001)	[209]
	-Experimental model: IL-2, TGF-β, TNF- α, IL23 with Naïve CD4+ T cell (In vitro) -Route of administration: Incubation with cell culture medium -Dosage: 20 μg/mL	-Decrease in IL-17 by 60% (*p* < 0.0001) -Decrease in TNF-α by 59.1% (*p* < 0.001) -Increase in FoxP3 by 150% (*p* < 0.001)	
	-Experimental model: 5% IMQ induced psoriasis in BALB/c mice (In vivo) -Route of administration: subcutaneous administration -Dosage: 5 mg/kg for 2 days	-Decrease in PASI by 70.67% (-) -Decrease in baker score by 73.17% (*p* < 0.0001) -Decrease in epidermal thickness by 71.42% (*p* < 0.0001) -Decrease in IL-23 by 56.25% (*p* < 0.01) -Decrease in IL-6 by 23.8% (*p* < 0.01) -Decrease in IL12 by 6.02% (*p* < 0.01) -Decrease in CD3+ T cells infiltration (-) -Decrease in Naïve CD4+ T cell by 65.2% (spleen) (*p* < 0.0001), 50% (LNs) (*p* < 0.01) -Decrease in Ly6G+ infiltration by 40% (*p* < 0.05) -Decrease in CD11c+ infiltration by 64.7% (*p* < 0.01) -Increase in CD206 by 110.5% (spleen) (*p* < 0.01, 70% (LNs)130% (*p* < 0.05) -Decrease in PD-1 by 55.8% (*p* < 0.0001) -Decrease in skin Ki67 by 50.4% (*p* < 0.001) -Decrease in skin PCNA by 46.5% (*p* < 0.0001) -Decrease in skin 4-HNE by 36.8% (*p* < 0.05) -Decrease in skin ACSL4 by 29.2% (*p* < 0.01) -Increase in skin GPX4 by 166.7% (n.s.)	
	-Experimental model: PMA, IL-4 with THP-1 cells (In vitro) -Route of administration: Incubation with cell culture medium -Dosage: -	-Decrease in CD86 by 42% (*p* < 0.01) -Decrease in NOS2 by 72% (*p* < 0.01) -Decrease in IL-6 by 82% (*p* < 0.01) -Increase in TGB1 by 240% (*p* < 0.001)	[194]
JAM -EV source cell: EVs were engineered by fusing EVs, isolated from ANXA1-overexpressing Jurkat or EL4 T cells, with M2 macrophage membranes -Cargo : Engineered EVs contained overexpressed ANXA1 protein and M2 macrophage membrane-derived proteins, including scavenger receptors such as CD163 and CD206 -Isolation method: EVs were isolated by ultracentrifugation, followed by co-extrusion with M2 macrophage membrane vesicles for engineering -Size/Markers: 144–169 nm/-	Experimental model : 5% IMQ induced psoriasis in C57BL/6 mice (In vivo) -Route of administration : subcutaneous administration -Dosage: 50 μg (Day 0/2/4/6)	-Normalization of splenic indices by 40–50% (*p* < 0.01) -Decrease PASI by 70–75% (*p* < 0.01) -Decrease in CD68 infiltration by 70–75% (*p* < 0.01) -Increase Arg1 by 150–200% (*p* < 0.01) -Decrease IL-1β by 70–75% (*p* < 0.01) -Decrease IL-6 by 70–75% (*p* < 0.01) -Decrease TNF-α by 70–75% (*p* < 0.01) -Increase in CD3+ T by 150–200% (*p* < 0.01) -Decrease IL-17A (-)	
	-Experimental model: TNF-α, IL-17A induced HaCaT cell (In vitro) -Route of administration: Incubation with cell culture medium -Dosage: 50 μg/mL	-Suppress NF-κB by 78% (*p* < 0.001) -Suppress Arg1 by 69% (*p* < 0.001) -Increase in PPP6C by 67% (*p* < 0.05) -Decrease in S100A8 by 61% (*p* < 0.001) -Decrease in S100A9 by 67% (*p* < 0.001) Decrease in CCL20 by 50% (*p* < 0.001) -Decrease in TNF-α by 47.6% (*p* < 0.01) -Decrease in IL-1 β by 47.6% (*p* < 0.01) -Decrease in CAMP by 55.9% (*p* < 0.001) -Decrease in CEBP/β by 26.7% (n.s.)	[211]
	-Experimental model: RNA + polyamine + peptide complex induced BMDCs (In vitro) -Route of administration: Incubation with cell culture medium -Dosage: 50 μg/mL	-Decrease in MHC II by 32.6% (*p* < 0.001) -Decrease in CD80+CD86+ by 24.2% (*p* < 0.001)	
	-Experimental model: Splenic cells from IMQ-induced psoriasis mice (In vitro) -Route of administration: Incubation with cell culture medium -Dosage: 50 μg/mL	-Decrease in IL-17A+CD4+ T cells by 60.6% (*p* < 0.05) -Decrease in IFN γ+CD4+ T cells by 54.1% (*p* < 0.01)	
nor@MSCs-EVs -EV source cell: UC-MSCs -Cargo: nor-NOHA -Isolation method: MSCs-EVs isolation (method unspecified), nor-NOHA loading via electroporation -Size/Markers: 67.7 nm/CD9, TSG101	-Experimental model: IMQ-induced psoriasis mouse model (In vivo) -Route of administration : IV administration -Dosage: 100 μg (Day 1/3/5)	-Decrease in epidermal thickness by 44.97% (*p* < 0.001) -Decrease in PASI by 66.67% (*p* < 0.001) -Decrease in Spleen Index by 25.92% (*p* < 0.05) -Decrease in Ki67+ by 43.82% (*p* < 0.001) -Decrease in PUT by 53.57% (epidermis), 53.85% (plasma) (*p* < 0.001) -Decrease in SPD by 53.8% (epidermis), 57.14 (plasma) (*p* < 0.001) -Decrease in Arg1 by 45.2% (*p* < 0.001) -Decrease in CD80+CD86+CD11c+ by 25.71% (skin) (*p* < 0.05), 48.61% (spleen) (*p* < 0.001), 61.88% (dLN) (*p* < 0.001) -Decrease in IFNγ+CD4+ T by 86.67% (skin) (*p* < 0.001), 43.1% (spleen) (*p* < 0.01), 50% (dLN) (*p* < 0.01) -Decrease in IL-17A+ CD4+ T by74.29% (skin) (*p* < 0.001), 11.1% (spleen) (*p* < 0.05), 75% (dLN) (*p* < 0.001)	
J@EV -EV source cell: UVB-induced HaCaT cells -Cargo: JPH203, IL-1RA -Isolation method: Differential centrifugation, Freeze-drying -Size/Markers: 150 nm	-Experimental model: IL-6, IL-1β induced HaCaT cells (In vitro) -Route of administration: Incubation with cell culture medium -Dosage: -	-Suppress in p-mTOR/mTOR by 66.6% (*p* < 0.001) -Suppress in p-NF-κB/NF-κB by 29.7% (*p* < 0.05) -Suppresses in HaCaT hyperproliferation by 25.5% (*p* < 0.05)	[212]
	-Experimental model: 5% IMQ induced psoriasis in BALB/c mice (In vivo) -Route of administration: subcutaneous injection -Dosage: 1.19 × 10^11^ particles for 5 days	-Decrease in PASI by 59% (-) -Decrease in p-NF-κB (-) -Decrease in IL-17A (-) -Decrease Ki67 (-) -Increase in Splenomegaly by 116.67% (*p* < 0.005) -Decrease in Th17 by 78.3% (*p* < 0.005)	
PTD2021P EVs ointment -EV source cell: MSCs -Cargo: -Isolation method: Conditioned medium was concentrated using a 100 kDa molecular weight cut-off membrane and subsequently filtered with a 0.22 nm filter. -Size/Markers: 50–200 nm/CD59, CD73, CD81, CD9, Alix	Experimental model : IMQ-induced mouse model (In vivo) -Route of administration : topical application -Dosage : 10× higher than the clinical dose, thrice a day (TID) for 20 days	-Decrease in C5b-9 -Decrease in IL-17 -Decrease in IL-23	[218]
	Experimental model : 10 healthy adult volunteers (Clinical activity) -Route of administration : topical application -Dosage: 70 mg MSCs EVs/gram of ointment, thrice a day (TID) for 20 days	-No marked adverse effect -Clinical Safety Laboratory Tests: Mean changes from baseline were minimal (*p* > 0.05) (e.g., Hematocrit:.44 ± 0.914% (*p* = 0.85), Glucose: 0.19 ± 1.309 mmol/L (*p* = 0.45), Alanine transaminase: 0.6 ± 8.85 U/L (*p* = 0.88)) -Inflammatory Blood Examinations: Mean changes from baseline were minimal (*p* > 0.05) (e.g., CRP: −0.11 ± 0.409 mg/L (*p* = 0.87), ESR: −2.0 ± 4.90 mm/h (*p* = 0.57)) -Vital Signs: Mean changes from baseline were minimal (*p* > 0.05) (e.g., Heart Rate: 2.3 ± 10.83 beats/min (*p* = 0.45), Systolic BP−4.3 ± 10.93 mmHg (*p* = 0.34)) -Local Skin Responses: 0% of subjects reported any local skin responses (mean VAS scores were 0.0 ± 0.00 for all symptoms)	
MSCs-sEVs-PD-L1 -EV source cell: Bone marrow-derived MSCs from C57BL/6 mice. -Cargo: PD-L1 -Isolation method: Differential Ultracentrifugation -Size/Markers: 100 nm/CD63, CD9, CD73	-Experimental model: LPS-induced bone marrow-derived macrophage (In vitro) -Route of administration: Incubation with cell culture medium -Dosage: -	-Decrease in CD80 by 69.52% (*p* < 0.001) -Increase in CD 206 by 100% (*p* < 0.001) -Decrease in IL-1β by 66.67% (*p* < 0.01) -Decrease in TNF-α by 60% (*p* < 0.05) -Increase in IL-10 by 100% (*p* < 0.05)	[179]
	-Experimental model: LPS-induced Bone Marrow-Derived Dendritic Cells (In vitro) -Route of administration : Incubation with cell culture medium -Dosage: -	-Decrease in CD80 by 62.45% (*p* < 0.001) -Decrease in CD86 by 52.76% (*p* < 0.01)	
	-Experimental model: Anti-CD3 and anti-CD28 antibody induced-Lymph node cells (In vitro) -Route of administration: Incubation with cell culture medium -Dosage: -	-Decrease in IFN-γ+ of CD4+T by 78.9% (*p* < 0.001) -Increase in Foxp3+ CD4+T by 516.87% (*p* < 0.001) -Decrease in Ki-67+CD3+ T by 81.69% (*p* < 0.001) - Increase in Annexin V+/PI+CD3+ T by 466.46% (*p* < 0.001) -Decrease in IFN-γ by 62.5% (*p* < 0.001) -Decrease in IL-2 by 75% (*p* < 0.01) -Increase in IL-4 by 100% (*p* < 0.05)	
	-Experimental model: 5% IMQ induced psoriasis C57BL/6 mice (In vivo) -Route of administration: IV injection via tail vein -Dosage: −50 μg for 4 days	-Decrease epidermal thickness by 55.5% (*p* < 0.001) -Decrease in CD45 by 46.6% (*p* < 0.001) -Decrease in CD3+ T infiltration by 50% (*p* < 0.001) -Decrease in CD4+ T infiltration by 50% (*p* < 0.001) Decrease in IFNγ+CD4+ T by 38.6% (*p* < 0.001) -Decrease in IL-17A+ CD4+ T by 59.2% (*p* < 0.001) -Increase in Fop3+CD4+ T by 76.5% (*p* < 0.001) -Decrease IL-17a mRNA by 66.7% (*p* < 0.001) -Decrease IFN-γ mRNA by 66.7% (*p* < 0.001) -Increase IL-4 mRNA by 200% (*p* < 0.001) -Decrease in CD11c+ infiltration by 66.7% (*p* < 0.001) -Decrease in CD80+CD86+CD11c+ by 61.5% (*p* < 0.001) -Decrease in F4/80+ infiltration by 75% (*p* < 0.001) -Decrease in CD80+ F4/80+ by 66.7% (*p* < 0.001) -Increase in CD206+ F4/80+ by 200% (*p* < 0.001) -Decrease IL-6 by 68% (*p* < 0.001) -Decrease TNF-α by 64.2% (*p* < 0.001) -Decrease IL-1β by 65.3% (*p* < 0.001)	

## Data Availability

No new data were created or analyzed in this study. Data sharing is not applicable to this article.

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
