# Peer review of "The Therapeutic Potential of Extracellular Vesicles in Psoriasis Treatment: Mechanisms, Applications, and Prospects"

_ijms, 2025, doi:10.3390/ijms262110297_

Round 1

Reviewer 1 Report

Comments and Suggestions for Authors

This manuscript focuses on the application of extracellular vesicles (EVs) in the treatment of psoriasis, and systematically investigates the pathological mechanisms of psoriasis, the therapeutic potential of EVs, and the related signalling pathways. The topic of the review is in line with the research hotspots at the intersection of regenerative medicine and dermatology, and is of great value in promoting the clinical translation of EVs in psoriasis. The review integrates a large amount of basic experimental and preliminary clinical data, and the logical framework is clear. However, there are still deficiencies in the in-depth explanation of the mechanism, the integration of research data, and the details of the clinical translation, which need to be further improved to support the persuasive conclusions.

Major comments:

1.The "Overview of psoriasis pathophysiology" section describes in detail the role of signalling pathways such as the IL-23/IL-17 axis, JAK/STAT and NF-κB, but does not establish a direct correlation between these pathways and the mechanism of action of EVs from different sources (e.g., MSC-EVs, Keratinocyte EVs). For example, it is mentioned that MSC-EVs can inhibit NF-κB activity, but it is not clear how specific molecules (e.g., specific miRNAs, proteins) carried by EVs can accurately regulate key nodes of the pathway (e.g., IKK complex, p65 nuclear translocation), and there is a lack of a complete chain of "EVs molecule cargo - signalling pathway target - pathology and phenotype improvement". There is a lack of analysis of the complete chain of "EVs molecular cargo - signalling pathway target - pathological phenotype improvement".

It is recommended to supplement the comparative analysis of the molecular mechanisms of EVs from different sources in regulating the core pathway, and to clarify the specificity and generality of the role of EVs by combining with specific research cases (e.g., which miRNAs are used by AD-MSC-EVs to down-regulate the expression of IL-23).

2.The section "EVs and psoriasis therapy" lists the results of studies of various EVs, but key experimental parameters (e.g., dose of EVs, route of administration, type of animal model) and efficacy indexes (e.g., decrease in PASI score, inhibition of inflammatory factors) have not been systematically integrated. For example, the dosage of UCMSC-EVs in different studies ranged from 50 μg to 200 μg, and the differences in efficacy were not analysed, making it difficult to form a unified conclusion on the dose-effect relationship.

It is suggested that a table be added to summarise the preparation methods, key characteristics (particle size, zeta potential, markers), experimental protocols and core efficacy data of EVs from different cellular sources, so as to facilitate readers to compare the strengths and weaknesses of different studies and the common patterns.  

3.The article mentions that the clinical application of EVs faces challenges such as isolation and purification, standardisation and delivery efficiency, but does not propose targeted solutions in the light of current technological advances. For example, in the large-scale production of EVs, only 3D culture is mentioned to increase the yield, without explaining the differences between this method and the traditional 2D culture in terms of EVs purity, bioactivity and cost; in the optimisation of the delivery system, there is a lack of quantitative data to support the increase in the targeting efficiency of engineered EVs (e.g. PD-L1-modified EVs).

It is recommended to supplement the comparative analysis of EVs production processes (e.g., ultracentrifugation, microfluidics), and discuss the key bottlenecks (e.g., batch stability, long-term safety) and potential breakthroughs of EVs from the laboratory to the clinic in the light of clinical studies (e.g., safety data of PTD2021P ointment in the Phase I trial) that have already been conducted.

4.In Figure 1, the cell of origin and target of key cytokines (e.g., IL-23, IL-17) were not labelled, and the inhibition of C5b-9 complex was not clearly demonstrated in Figure 3; and in Table 1 (Summary of the therapeutic potential of EVs), some of the studies were too brief (e.g., only mentioning "reducing inflammatory factors"), lacking specific values and statistically significant analysis. In Table 1, the description of "in vitro/in vivo activity" in some studies is too brief (e.g., only mentioning "reduction of inflammatory factors"), and there is a lack of specific values and statistical significance analysis.

It is recommended to improve the annotations of the charts, add the relationship of key molecules and specific details of the experimental data (e.g., "IL-17 expression was reduced by 40%, P<0.01"), and ensure that the information in the charts is consistent with the descriptions in the text.

5.There are inconsistencies in the expression of some terms in the text, such as "mesenchymal stem cells" is sometimes abbreviated as "MSCs" and sometimes as "MSCs"; "psoriasis area severity index" is abbreviated as "PASI" and "PSI" in different chapters, which may cause confusion. area severity index" is abbreviated as "PASI" and "PSI" in different chapters, which is easily confused.

Author Response

Thank you sincerely for your careful/detailed review. Please see the attachment.

Reviewer 2 Report

Comments and Suggestions for Authors

The manuscript provides a comprehensive and timely review of extracellular vesicles (EVs) as emerging therapeutic agents for psoriasis. The authors successfully integrate knowledge of psoriasis pathophysiology with the mechanistic basis of EV therapy, highlighting preclinical evidence, early clinical applications, and future perspectives. The topic is highly relevant given the limitations of current biologics and the rising interest in cell-free regenerative strategies.

However, there are several areas that require clarification, refinement, and further contextualization to enhance its scientific rigor and clinical relevance.

1.  To strengthen the introduction and provide the background, the following references should be included:
Chen HH, Abed SR. Update Aetiopathogenesis and Treatment of Psoriasis: A Literature Review. J Dermatol Res. 2023;4(1):1-13. doi: https://doi.org/10.46889/JDR.2023.4201

2. Consider condensing redundant sections on immune mechanisms (e.g., IL-23/IL-17 axis, JAK/STAT pathway) and emphasizing how these pathways are specifically modulated by EVs.

3. Please provide a clearer conceptual framework that distinguishes between (a) EVs as natural immunomodulators, (b) engineered/modified EVs, and (c) EVs as drug delivery systems.

4. Please discuss which EV sources (MSC-derived vs. immune cell-derived vs. keratinocyte-derived) show the most robust evidence. Also discuss if there are differences in therapeutic mechanisms between stem cell–EVs and non–stem cell EVs.

5. Figures explaining psoriasis pathogenesis are detailed but reproduced from prior sources. To improve originality, consider adding a new schematic summarizing how EVs intervene at specific checkpoints in the psoriasis inflammatory cascade.

6. It would be valuable to discuss future potential for combining EVs with biologics or small molecules. Also, the safety considerations (e.g., risk of unwanted immunosuppression or off-target effects) should be addressed.

Author Response

(The authors gave the same response as above.)

Round 2

Reviewer 1 Report

Comments and Suggestions for Authors

he author has addressed the key issues with commendable clarity, and I recommend accepting this article.

Reviewer 2 Report

Comments and Suggestions for Authors

The manuscript has been sufficiently improved to warrant publication in IJMS. No further revision is suggested.